# Enterovirus A71 3AB protein facilitates immune evasion by blocking cGAS recognition of mtDNA

Peng Sun [1,2,3,6]✉, Xinya Yang[1,6], Jing Cui[1], Guicun Fang[4], Yuqin Wu[1], Jingyi Chang[1], Xiaofei Li[1], Yinli Xie[1,2], Lipeng Gan[1,2], Lina Ma[1] & Zhiyong Li [1,2,3,5]✉

## Abstract

**Mitochondrial DNA (mtDNA) is a powerful stimulator of the innate immune system and has been shown to trigger cytosolic DNA-sensing signaling during picornavirus infection. In this study, we observe that EV-A71 infection induces mitochondrial damage and leads to the release of mtDNA into the cytoplasm, which was mediated by the viral 2B protein. Despite this release, EV-A71 effectively suppresses the cGAS–STING-mediated type I interferon (IFN-I) response. We identify the nonstructural protein 3AB as a key viral antagonist of mtDNA sensing. Mechanistically, 3AB directly binds cytosolic mtDNA and disrupts cGAS–DNA phase separation, thereby suppressing cGAS–STING-dependent antiviral signaling. The immunosuppressive function of 3AB depends on the "3B + 7" region, with mutations impairing its mtDNA binding and IFN-I suppression. Moreover, the 3AB proteins from coxsackievirus A9 (CVA9) and A16 (CVA16) also exhibit mtDNA-binding ability. This study reveals a novel immune evasion strategy by blocking mtDNA-triggered immune signaling, providing new insights into the interplay between viral infection and mitochondrial immune defense.**

**Keywords** cGAS; EV-A71; Immune Evasion; mtDNA; Phase Separation
**Subject Categories** Microbiology, Virology & Host Pathogen Interaction; Organelles; Signal Transduction

## Introduction

Mitochondria, often regarded as powerhouses of immunity, play a central role in regulating diverse cellular activities and coordinating host responses to viral infections (Mills et al, 2017). Mitochondria serve as critical platforms for immune signaling (West et al, 2011). Mitochondrial antiviral signaling protein (MAVS), anchored on the outer mitochondrial membrane, is activated by the viral RNA sensors RIG-I and MDA5, triggering downstream antiviral immune responses (Sorouri et al, 2022). Numerous studies have demonstrated that mitochondrial DNA (mtDNA) is a powerful stimulator of the innate immune system (Riley and Tait, 2020; West and Shadel, 2017). Mammalian mtDNA is a small, circular double-stranded DNA molecule of approximately 16.5 kilobases that encodes thirteen essential subunits of the mitochondrial OXPHOS system, as well as two rRNAs and 22 tRNAs required for mitochondrial protein synthesis (Gustafsson et al, 2016). Mitochondrial transcription factor A (TFAM) tightly compacts mtDNA into DNA-protein structures known as mitochondrial nucleoids, which reside within the mitochondrial matrix (Kang et al, 2007; Kanki et al, 2004). Similar to chromatin in the nucleus, mitochondrial nucleoids play a critical role in coordinating and regulating essential processes, including mtDNA replication, repair, transcription, and gene expression (Tan et al, 2024).

Under conditions of cellular stress or during viral infections (conditions that cause mitochondrial dysfunction), mitochondrial damage can lead to the release of mtDNA into the cytosol (De Gaetano et al, 2021). The permeabilization of the mitochondrial outer membrane is essential for the release of mtDNA (Flores-Romero et al, 2023). During apoptosis, activated BAK and BAX oligomerize on the mitochondrial outer membrane, where they form large pores. These macropores facilitate the herniation of the inner membrane, allowing mitochondrial matrix components, including mtDNA, to enter the cytoplasm as inner membrane regions lose their integrity (Cosentino et al, 2022; McArthur et al, 2018). Under oxidative stress, mitochondria release short mtDNA fragments through pores formed by voltage-dependent anion channel (VDAC) oligomers in the outer mitochondrial membrane. The inhibition of VDAC oligomerization by VBIT-4 significantly reduces mtDNA release and suppresses the activation of interferon (IFN) signaling (Kim et al, 2019; Yan et al, 2020). TFAM deficiency significantly disrupts mtDNA packaging, organization, and distribution, leading to its release into the cytosol and subsequent activation of innate immune signaling (West et al, 2015). Research has demonstrated that viroporins, transmembrane pore-forming proteins derived from viruses such as the influenza virus M2, can facilitate the release of mtDNA into the cytosol (Moriyama et al, 2019). Similarly, picornavirus infection induces mitochondrial dysfunction, leading to mtDNA release through the mitochondrial permeability transition pore (mPTP) (Liu et al, 2023). Released mtDNA serves as a damage-associated molecular pattern (DAMP)

[1]School of Basic Medical Sciences, Wenzhou Medical University, Wenzhou, China. [2]Cixi Biomedical Research Institute, Wenzhou Medical University, Ningbo, China. [3]Institute of Virology, Wenzhou Medical University, Wenzhou, China. [4]Microscopy Core Facility, Westlake University, Hangzhou, China. [5]Institute of Biology, Hebei Academy of Sciences, Shijiazhuang, China. [6]These authors contributed equally: Peng Sun, Xinya Yang. ✉E-mail: sunpeng@wmu.edu.cn; lizhiyong@wmu.edu.cn

and is recognized as foreign by innate immune receptors, especially the DNA sensor cyclic GMP-AMP synthase (cGAS) (Aarreberg et al, 2019). Upon binding to mtDNA, cGAS catalyzes the synthesis of the second messenger cyclic GMP-AMP (cGAMP). cGAMP binds to and activates STING on the endoplasmic reticulum, which recruits TBK1 to phosphorylate IRF3, facilitating its dimerization and translocation to the nucleus. In the nucleus, IRF3 induces the expression of type I interferon (IFN-I) and other interferon-stimulated genes, triggering a robust antiviral immune response (Chen et al, 2016; Hopfner and Hornung, 2020).

Hand, foot and mouth disease (HFMD) is a highly contagious viral infection affecting young children under 5 years of age worldwide (Koh et al, 2016; Zhu et al, 2023). HFMD is a febrile illness characterized by a maculopapular rash or blisters on the hands, feet, groin, and buttocks and is associated with painful ulcerative lesions of the mouth (Aswathyraj et al, 2016; Esposito and Principi, 2018). EV-A71, a member of the Picornaviridae family, is a major causative agent of HFMD, and in severe cases, it can lead to neurological complications such as encephalitis (Wang and Liu, 2009; Wong et al, 2010; Xing et al, 2022). The relationship between EV-A71 and the host immune response is multifaceted, particularly with respect to the interactions between EV-A71 infection and the host's mitochondria and mtDNA, which remain unclear.

In our experiments, we found that despite the release of mtDNA into the cytoplasm during EV-A71 infection, the virus suppressed the cGAS–STING-mediated IFN-I response. Specifically, we demonstrated that the EV-A71 nonstructural protein 3AB binds cytoplasmic mtDNA, thereby preventing its recognition by cGAS, disrupting cGAS–DNA phase separation, and suppressing the antiviral immune response. In addition, 3AB proteins from coxsackievirus A9 (CVA9) and A16 (CVA16) were also capable of binding mtDNA. Our findings revealed a novel immune evasion strategy of EV-A71, in which the virus blocks the mtDNA–cGAS–STING pathway, effectively suppressing the host antiviral response. Our study provides new insights into the molecular interplay between EV-A71 and the host's mitochondrial immune defense mechanisms, highlighting potential targets for therapeutic intervention.

## Results

### EV-A71 infection induces mtDNA release

To assess the impact of EV-A71 infection on mitochondrial function, mitochondrial ultrastructural alterations were analyzed by transmission electron microscopy (TEM). We found that EV-A71 infection led to noticeable changes in mitochondrial morphology, such as mitochondrial membrane disruption and the disappearance of mitochondrial cristae (Fig. 1A). Next, we characterized the damage to mitochondria caused by EV-A71 infection by assessing the mitochondrial membrane potential (Δψm), a critical indicator of mitochondrial activity, using the JC-10 assay. In healthy mitochondria, JC-10 forms red-fluorescent aggregates, whereas in depolarized mitochondria it remains as green-fluorescent monomers, with a reduced red/green fluorescence ratio indicating mitochondrial depolarization (Sundaram et al, 2022). Carbonyl cyanide m-chlorophenyl hydrazone (CCCP), which is an uncoupler that eliminates the mitochondrial membrane potential, was used as a positive control (Kamer et al, 2018). As shown in Fig. 1B, EV-A71 infection significantly disrupted the mitochondrial membrane

potential. To further clarify whether mitochondrial damage resulting from EV-A71 infection leads to the release of mtDNA, we conducted an analysis of mtDNA in the cytoplasm. The cytosolic fractions were isolated (Fig. EV1A–D). We found that EV-A71 infection led to the release of mtDNA into the cytoplasm in THP-1, RD, U251 and HeLa cells (Fig. 1C), which is consistent with previous study (Liu et al, 2023). These results indicate that EV-A71 infection can damage host cell mitochondria, leading to the release of mtDNA into the cytoplasm.

To determine whether EV-A71-induced mitochondrial damage and mtDNA release require active viral replication, we treated host cells with UV-inactivated EV-A71. The loss of infectivity was confirmed by viral titer assay. Notably, UV-inactivated virus, which retains structural proteins, failed to induce mtDNA release (Fig. EV1E), suggesting that active replication is essential for this process. To identify the viral protein responsible, we constructed expression vectors for EV-A71 nonstructural proteins (Fig. 1D). Given that 3B is only 22 amino acids long, we expressed its precursor 3AB for better characterization, while excluding 2A protease (2Apro) from further analysis due to its strong cytotoxicity. Among them, we found that 2B markedly promoted mtDNA release in HeLa cells (Fig. 1E). Consistently, overexpression of 2B protein resulted in pronounced mitochondrial damage and a reduction in mitochondrial membrane potential (Fig. EV1F,G). And 2B protein activated the TBK1–IRF3 signaling pathway through the release of mtDNA (Fig. EV1H). As an ion channel protein (viroporin), EV-A71 2B protein is consistent with prior reports showing that viroporins from other viruses, such as the M2 protein of Influenza A virus and the 2B protein of encephalomyocarditis virus (EMCV) can induce mitochondrial DNA release (Moriyama et al, 2019).

### EV-A71 suppresses the mtDNA–cGAS–STING pathway to evade host immunity

Upon viral infection, mtDNA is released into the cytoplasm, where it is sensed by cGAS, triggering the cGAS–STING pathway and inducing IFN-I production (Hu and Shu, 2020). We then evaluated the role of mtDNA in the antiviral immune response. Although EV-A71 infection resulted in mtDNA release into the cytosol, it did not effectively trigger IFN-β production (Fig. 2A). Similarly, EV-A71 infection also failed to induce IFN-β production in HeLa and THP-1 cells (Fig. 2B,C), indicating that EV-A71 blocks the mtDNA-mediated antiviral immune pathway. Subsequently, we examined whether EV-A71 infection could suppress responses to the cGAS activator poly(dA:dT), a widely used dsDNA mimetic. Similarly, we found that EV-A71 inhibited poly(dA:dT)-induced IFN-I responses in both HeLa and U251 cells (Fig. 2D,E). Pseudorabies virus (PRV), a double-stranded virus, activated the IFN-I response by triggering the cGAS–STING pathway (Wang et al, 2015). EV-A71 also suppressed the IFN-I response induced by PRV infection in HeLa and THP-1 cells (Fig. EV2A,B). These results suggest that while EV-A71 disrupts mitochondrial integrity, it efficiently blocks the mtDNA–cGAS–STING pathway, preventing the downstream activation of IFN-I signaling. We further investigated whether EV-A71 interfered with immune responses activated by endogenous mtDNA by generating TFAM knockout (KO) cell lines. TFAM is essential for mtDNA packaging within nucleoids, and its loss leads to mtDNA leakage, which activates the

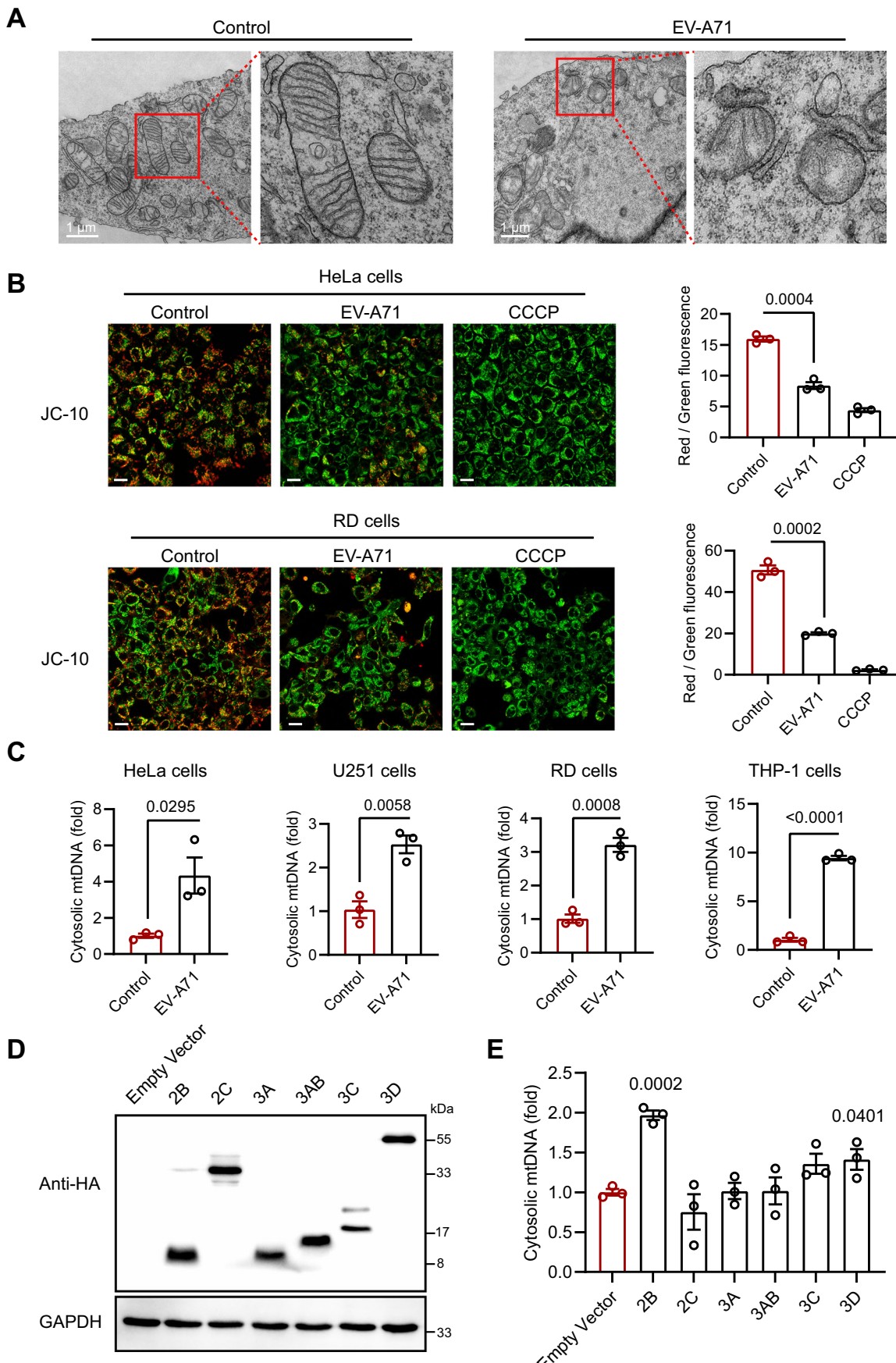

Figure 1. EV-A71 infection disrupts mitochondria and results in mtDNA release.

(A) Mitochondrial morphology analyzed in RD cells infected with EV-A71. After 12 h of EV-A71 infection (MOI = 1) in RD cells, mitochondrial morphology was examined by transmission electron microscopy. Scale bars, 1 µm. (B) After infecting HeLa or RD cells with EV-A71 (MOI = 1) for 24 or 18 h, respectively, the mitochondrial membrane potential (ΔΨm) was assessed with JC-10, with CCCP serving as a positive control. In healthy mitochondria, JC-10 accumulates in the mitochondrial matrix and forms aggregates that emit red fluorescence (Ex/Em = 540/590 nm). In contrast, in depolarized or damaged mitochondria, JC-10 remains in its monomeric form in the cytoplasm, emitting green fluorescence (Ex/Em = 490/525 nm). The ratio of red to green fluorescence serves as an indicator of mitochondrial membrane potential. Images were examined by Nikon C2 confocal microscopy (left panel). Scale bars, 20 µm. The ratio of red to green fluorescence for JC-10 was analyzed with a multifunctional microplate reader (right panel). Data were presented as mean ± SEM, $n = 3$ biological replicates. Unpaired $t$ test was used for statistical analysis. (C) HeLa, U251, RD, and THP-1 cells were infected with EV-A71 (MOI = 1) for 12, 12, 6, and 24 h, respectively. Cytosolic fractions were isolated, and qPCR was conducted to measure the cytosolic mtDNA. Data were presented as mean ± SEM, $n = 3$ biological replicates. Unpaired $t$ test was used for statistical analysis. (D, E) HeLa cells were transfected with EV-A71 nonstructural proteins for 24 h. (D) Western blotting analysis of EV-A71 nonstructural proteins was performed. (E) Cytosolic fractions were isolated, and qPCR was conducted to measure the cytosolic mtDNA. Data were presented as mean ± SEM, $n = 3$ biological replicates. Unpaired $t$ test was used for statistical analysis. The experiments were repeated at least three times with the similar results Source data are available online for this figure.

cGAS–STING–IRF3 pathway to promote IFN-I expression (West et al, 2015). As expected, TFAM KO HeLa cells presented increased mtDNA leakage and interferon expression, and reduced EV-A71 infection (Figs. 2F,G and EV2C,D). Treatment with the cGAS antagonist RU.521 or the STING antagonist H151 significantly suppressed mtDNA-induced IFN-β production as well as STING and TBK1 phosphorylation in TFAM-KO cells, confirming that the enhanced interferon response is dependent on cGAS–STING signaling (Fig. EV2E,F). EV-A71 infection inhibited TFAM deficiency-induced interferon expression and TBK1–IRF3 pathway activation in TFAM KO cells (Fig. 2H,I). These findings suggest that EV-A71 can effectively inhibit the immune response triggered by TFAM deficiency and block the activation of cGAS–STING signaling.

We subsequently investigated the mechanism by which EV-A71 evades the mtDNA–cGAS–STING pathway. Initially, we assessed the impact of EV-A71 infection on the expression levels of key proteins involved in this pathway. Our results demonstrated that EV-A71 infection did not significantly alter the expression of cGAS, STING, TBK1, or IRF3, indicating that the virus did not evade the cGAS–STING pathway through the degradation of these components (Fig. 3A). Next, we evaluated the production of 2'3'-cGAMP, a marker of cGAS activation (Zhang et al, 2014a). ELISA analysis showed that EV-A71 infection alone did not significantly activate 2'3'-cGAMP production. Notably, EV-A71 infection markedly suppressed poly(-dA:dT)-induced 2'3'-cGAMP production (Fig. 3B). These findings indicate that EV-A71 interfere with cGAS activity, preventing efficient activation of cGAS by cytosolic mtDNA.

## EV-A71 3AB suppresses the cGAS–STING pathway

To determine whether EV-A71-mediated inhibition of the cGAS pathway required viral replication, we performed UV inactivation of EV-A71. UV-inactivated EV-A71 did not inhibit 2'3'-cGAMP production and IFN-β expression induced by poly(dA:dT) in HeLa cells (Fig. 3B,C), which endogenously express cGAS and STING, indicating that viral replication is necessary for EV-A71 to inhibit the cGAS pathway. Since 293 T cells do not express cGAS or STING, we transiently transfected cGAS and STING into these cells (Sun et al, 2013; Zhang et al, 2014b). Luciferase activity assays further demonstrated that UV-inactivated EV-A71 failed to significantly suppress the IFN-I response activated by poly(dA:dT) (Figs. 3D and EV3A). Given that UV-inactivated EV-A71 retains structural proteins, we subsequently investigated the potential role of nonstructural

proteins in modulating the cGAS–STING signaling pathway. Luciferase activity assays demonstrated that nonstructural protein 3AB significantly inhibited the IFN-I response activated by poly(dA:dT) in 293T cells (Figs. 3E and EV3B). In addition, we found that the EV-A71 2C protein also inhibited the IFN-I response, consistent with previous reports (Liu et al, 2023). Similarly, in HeLa cells, we observed the same inhibitory effect of EV-A71 3AB on the cGAS–STING pathway (Fig. EV3C), suggesting that EV-A71 3AB can suppress the cGAS–STING pathway.

Next, we used another cGAS agonist, HT-DNA, and found that 3AB markedly suppressed HT-DNA-induced IFN-β expression at both the mRNA and protein levels in HeLa cells, as determined by qRT-PCR and ELISA (Fig. 3F,G). Consistent with these results, in MG63 cells, 3AB significantly inhibited poly(dA:dT)-induced and HT-DNA-induced IFN-β production as well as CXCL10/IP-10 expression (Figs. 3H–J and EV3D). In contrast, 3AB did not significantly inhibit IFN-β expression induced by poly(I:C), a RIG-I/MAVS agonist (Fig. EV3E–H). Furthermore, the 3AB protein inhibited the TKB1–IRF3 signaling pathway (Fig. EV3I) and IFN-β expression activated by endogenous mtDNA in TFAM-KO cells (Fig. 3K). At 6 h post-infection, the 3AB protein was significantly expressed, and its expression level progressively increased throughout the infection (Fig. 3L). Collectively, these findings demonstrate that 3AB is capable of suppressing mtDNA-mediated IFN-I responses.

## EV-A71 3AB inhibits cGAS recognition of mtDNA and phase separation

Consistent with observations during viral infection, 3AB expression did not significantly alter cGAS or STING protein levels, yet markedly suppressed poly(dA:dT)-induced 2'3'-cGAMP production (Figs. 4A and EV4A). To further determine whether 3AB acts upstream or downstream of STING, cells were treated with the STING agonists cGAMP or diABZI. In contrast to its inhibitory effect on DNA-triggered signaling, 3AB did not suppress 2'3'-cGAMP- or diABZI-induced activation of downstream STING signaling pathways, nor the expression of IFN-β and CXCL10 (Fig. EV4B–F). These results indicate that 3AB might target cGAS, rather than STING, to inhibit DNA sensing.

We further explored the potential of 3AB to inhibit cGAS activity through direct interaction. Co-expression of human cGAS with EV-A71 3AB revealed no significant interaction between the two proteins (Fig. 4B,C). Next, we tested whether 3AB blocks the recognition of mtDNA by cGAS. To assess this, we labeled mtDNA with

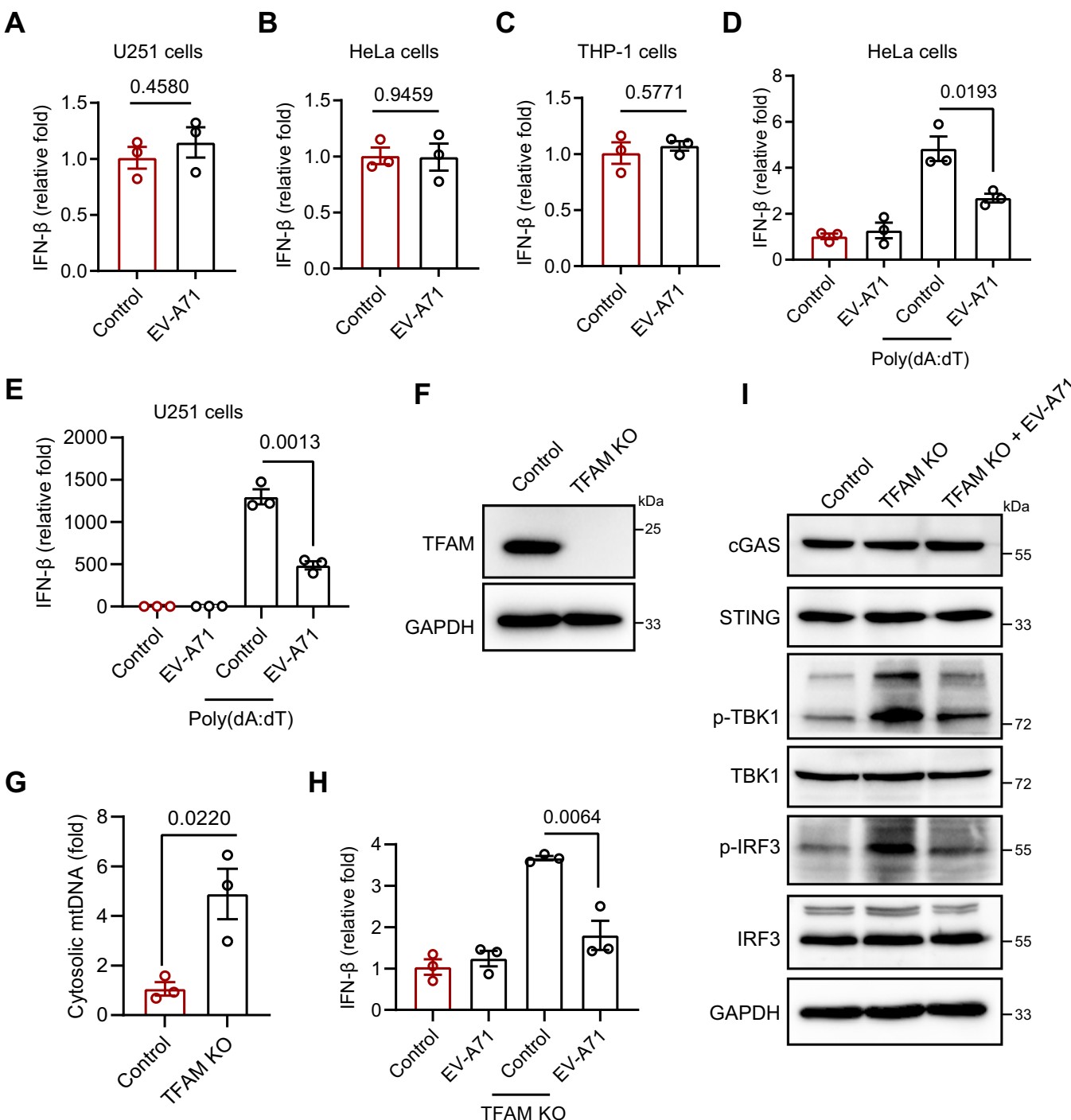

**Figure 2. EV-A71 infection modulates the mtDNA–cGAS–STING pathway.**

(A–C) HeLa, U251, and THP-1 cells were infected with EV-A71 at an MOI of 1 for 12 or 24 h. The mRNA expression of IFN-β was assessed by qRT–PCR. Data were presented as mean ± SEM, n = 3 biological replicates. Unpaired t test was used for statistical analysis. (D, E) HeLa (D) and U251 (E) cells were transfected with 100 ng/mL poly(dA:dT) for 6 h, followed by EV-A71 infection (MOI = 1) for 12 h. qRT–PCR was then performed to measure the expression of IFN-β mRNA. Data were presented as mean ± SEM, n = 3 biological replicates. Unpaired t test was used for statistical analysis. (F–I) EV-A71 infection (MOI = 1) impaired the activation of the IFN-I pathway in TFAM-knockout cells. (F) sgRNA-mediated knockout of TFAM was assessed by Western blotting with an anti-TFAM antibody. (G) Cytosolic mtDNA levels were quantified by qPCR in TFAM knockout and mock-treated cells. (H, I) TFAM knockout and mock-treated cells were infected with EV-A71 for 12 h. IFN-β mRNA expression was measured by qRT–PCR, and protein levels of cGAS, STING, as well as total and phosphorylated TBK1 and IRF3 were analyzed by western blotting. Data were presented as mean ± SEM, n = 3 biological replicates. Unpaired t test was used for statistical analysis. The experiments were repeated at least three times with the similar results. Source data are available online for this figure.

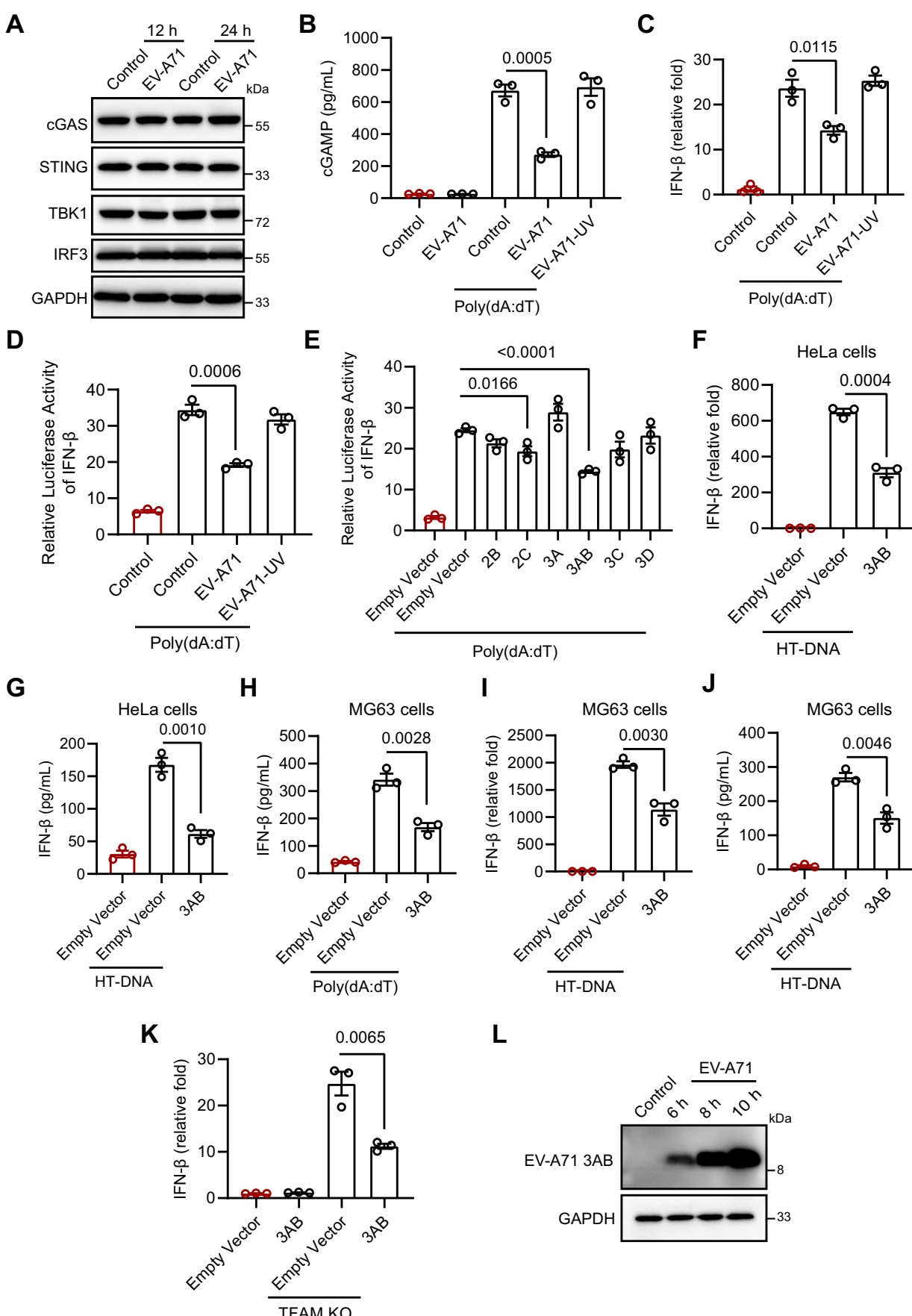

Figure 3. EV-A71 3AB modulates the cGAS–STING pathway.

(A) After 12 and 24 h of EV-A71 infection (MOI = 1), Western blotting analysis was performed to assess total cGAS, STING, TBK1, and IRF3 levels in HeLa cells. (B) HeLa cells were transfected with 100 ng/mL poly(dA:dT) for 6 h, followed by EV-A71 or EV-A71-UV (MOI = 1) infection for 12 h. Intracellular cGAMP levels were measured by ELISA. Data were presented as mean ± SEM, n = 3 biological replicates. Unpaired t test was used for statistical analysis. (C) HeLa cells were transfected with 100 ng/mL poly(dA:dT) for 6 h, followed by EV-A71 or EV-A71-UV (MOI = 1) infection for 12 h. qRT–PCR was then performed to measure the expression of IFN-β mRNA. Data were presented as mean ± SEM, n = 3 biological replicates. Unpaired t test was used for statistical analysis. (D) cGAS, STING, poly(dA:dT), and IFN-β luciferase reporter plasmids were transfected into 293 T cells, which were then infected with EV-A71 or EV-A71-UV (MOI = 1) for 12 h. The IFN-β luciferase activity was measured with a dual-luciferase reporter assay. Data were presented as mean ± SEM, n = 3 biological replicates. Unpaired t test was used for statistical analysis. (E) cGAS, STING, poly(dA:dT), and IFN-β luciferase reporter plasmids were transfected into 293 T cells, which were then transfected with plasmids of EV-A71 nonstructural proteins for 24 h. The IFN-β luciferase activity was measured with a dual-luciferase reporter assay. Data were presented as mean ± SEM, n = 3 biological replicates. Unpaired t test was used for statistical analysis. (F, G) HeLa cells were transfected with EV-A71 3AB plasmid for 24 h, followed by transfection with 500 ng/mL HT-DNA for 12 h. (F) IFN-β mRNA expression was measured by qRT–PCR. (G) IFN-β protein expression in supernatant was determined by ELISA. Data were presented as mean ± SEM, n = 3 biological replicates. Unpaired t test was used for statistical analysis. (H) MG63 cells were transfected with 100 ng/mL poly(dA:dT) for 6 h, followed by transfection with EV-A71 3AB plasmid for 18 h. IFN-β protein expression in supernatant was determined by ELISA. Data were presented as mean ± SEM, n = 3 biological replicates. Unpaired t test was used for statistical analysis. (I, J) MG63 cells were transfected with EV-A71 3AB plasmid for 18 h, followed by transfection with 500 ng/mL HT-DNA for 6 h. (I) IFN-β mRNA expression was measured by qRT–PCR. (J) IFN-β protein expression in supernatant was determined by ELISA. Data were presented as mean ± SEM, n = 3 biological replicates. Unpaired t test was used for statistical analysis. (K) The 3AB plasmid was transfected into TFAM knockout and mock-treated cells for 24 h, after which IFN-β mRNA expression was measured by qRT–PCR. Data were presented as mean ± SEM, n = 3 biological replicates. Unpaired t test was used for statistical analysis. (L) RD cells were infected with EV-A71 at an MOI of 1 for 6, 8 and 10 h, western blotting analysis was performed to assess EV-A71 3AB. The experiments were repeated at least three times with the similar results. Source data are available online for this figure.

bromodeoxyuridine (BrdU) and co-transfected HeLa cells with cGAS-Myc expression constructs and BrdU-labeled mtDNA, in the presence or absence of 3AB. After transfection, we pulled down cGAS-Myc using an anti-Myc antibody, and the immunoprecipitated complexes were analyzed by dot blotting. Immunoprecipitation studies revealed that mtDNA efficiently co-precipitated with cGAS. However, in the presence of EV-A71 3AB, the quantity of mtDNA precipitated with cGAS significantly decreased (Fig. 4D). Consistent with the dot blotting results, the results of the qPCR analyses indicated that the interaction between cGAS and mtDNA was disrupted by 3AB (Fig. 4E). Given that DNA-induced liquid–liquid phase separation of cGAS is critical for its enzymatic activity and downstream signaling (Xu et al, 2021; Zhou et al, 2021), we next examined whether EV-A71 3AB could restrict the formation of cGAS–DNA condensates. Overexpression of 3AB did not affect cGAS protein levels (Fig. EV4G). However, 3AB overexpression markedly impaired cGAS condensate formation (Fig. 4F). Quantitative analysis confirmed a significant reduction in the size of cGAS condensates in the presence of 3AB (Fig. 4G). To further investigate this effect, we expressed and purified recombinant cGAS and 3 AB proteins for in vitro phase separation assays (Fig. EV4H). We found that cGAS alone did not undergo phase separation in the absence of DNA, whereas the addition of DNA robustly induced cGAS condensate formation (Fig. EV4I). Intriguingly, inclusion of purified 3 AB protein disrupted the phase-separated cGAS–DNA condensates (Fig. 4H). Importantly, in vitro enzymatic activity assays showed that purified 3AB inhibited cGAS enzymatic activity in a dose-dependent manner (Fig. 4I). Collectively, these findings suggest that 3AB disrupts the recognition of mtDNA by cGAS, thereby interfering with cGAS–DNA phase separation and ultimately suppressing the antiviral immune response.

## EV-A71 3AB binds cytosolic mtDNA to inhibit the cGAS–STING pathway

Previous research demonstrated that EV-A71 3AB binds nucleic acids, acting as both a helicase and a molecular chaperone (Tang et al, 2014). On the basis of these findings, we hypothesized that 3AB may engage with cytosolic mtDNA, thereby obstructing cGAS

recognition of mtDNA. To test this hypothesis, we isolated mitochondria and subsequently purified mtDNA from HeLa cells. The purified mtDNA and recombinant HA-tagged 3AB plasmid were cotransiently transfected into HeLa cells. As controls, cells were transfected with an HA-empty vector or HA-tagged 3A, a viral protein that does not impact the IFN-I pathway. HA-tagged proteins were subsequently immunoprecipitated using an anti-HA antibody, and the associated mtDNA was examined by qPCR. The results showed that 3AB efficiently precipitated mtDNA, whereas 3A did not (Fig. 5A). Next, we labeled mtDNA with BrdU and co-transfected HeLa cells with a 3AB-HA expression construct and purified BrdU-mtDNA. After transfection, we pulled down 3AB-HA, and the immunoprecipitated complexes were analyzed by dot blotting, which revealed a distinct BrdU signal, confirming the potential interaction between 3AB and mtDNA (Fig. 5B). Additionally, confocal microscopy demonstrated the colocalization of EV-A71 3AB with EdU-labeled mtDNA, further supporting the association between 3AB and mitochondrial DNA (Fig. 5C). Similarly, in TFAM-deficient cells, the EV-A71 3AB protein also bound endogenous mtDNA, further supporting the notion that 3AB engages with cytosolic mtDNA (Fig. 5D). To further determine whether the 3AB-mtDNA interaction occurs under physiological conditions, we next investigated whether this association occurs during viral infection. HeLa cells were infected with EV-A71 for 12 h, followed by immunoprecipitation using a 3AB-specific antibody. The immunoprecipitated complexes were analyzed by qPCR to detect endogenous mtDNA. As shown in Fig. 5E,F, mtDNA was specifically enriched in the 3AB immuno-precipitates, indicating that 3AB interacts with endogenous mtDNA during EV-A71 infection. To further validate this interaction, we quantitatively analyzed the binding of 3AB to DNA at different concentrations using surface plasmon resonance (SPR), which confirmed a direct and dose-dependent 3AB–DNA interaction (Fig. 5G,H). These results demonstrate that EV-A71 3AB specifically associated with mtDNA.

EV-A71 3AB exhibits nucleic acid chaperone activity essential for viral RNA replication. This function is attributed to specific regions and amino acids within the protein (Tang et al, 2014). The

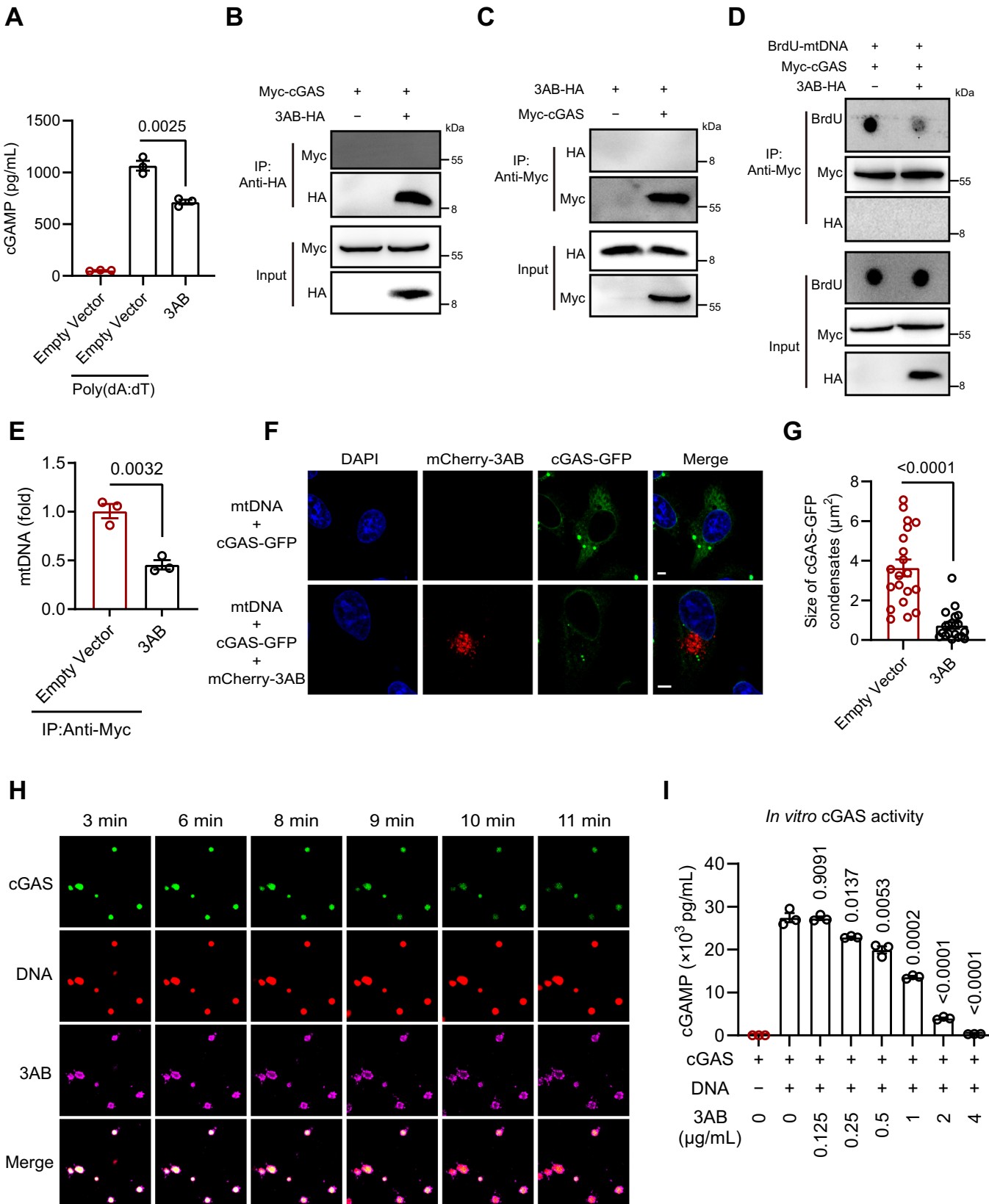

**Figure 4. EV-A71 3AB sequesters mtDNA from the cGAS-bound state.**

(A) HeLa cells were transfected with 100 ng/mL poly(dA:dT) for 6 h, followed by transfection with EV-A71 3AB plasmid for 24 h. Intracellular cGAMP levels were measured by ELISA. Data were presented as mean ± SEM, n = 3 biological replicates. Unpaired t test was used for statistical analysis. (B, C) Investigation of the interaction between 3AB and cGAS by co-IP. 3AB and cGAS plasmids were transfected into 293 T cells, and cell lysates were immunoprecipitated with either anti-HA or anti-Myc antibodies, followed by detection with the respective other antibodies. (D, E) 3AB impaired the binding between cGAS and mtDNA. After 24 h of co-transfection with 3AB, cGAS plasmids and mtDNA, the cytosol was isolated, followed by immunoprecipitation with anti-Myc (D). mtDNA was assessed by dot blotting (D) or qPCR (E). Data were presented as mean ± SEM, n = 3 biological replicates. Unpaired t test was used for statistical analysis. (F, G) HeLa cells were co-transfected with cGAS-GFP, mCherry-3AB plasmids and mtDNA for 24 h. (F) Representative images of cGAS-GFP and mCherry-3AB were examined with confocal microscope. Scale bars, 5 μm. (G) The size of GFP-cGAS condensates were measured from n = 20 condensates. Data were presented as mean ± SEM, n = 3 biological replicates. Unpaired t test was used for statistical analysis. (H) Recombinant cGAS (20 μM) and 100-bp dsDNA (10 μM) were incubated for 10 min, followed by the addition of 3AB (20 μM). Time-lapse imaging was performed using a Leica STELLARIS 5 confocal microscope. Scale bars, 5 μm. (I) In vitro cGAS activity was assessed by measuring cGAMP production in the presence of 100-bp DNA and cGAS with different concentrations of 3AB. Data were presented as mean ± SEM, n = 3 biological replicates. Unpaired t test was used for statistical analysis. The experiments were repeated at least three times with the similar results. Source data are available online for this figure.

chaperone activity is mediated by the "3B + 7" region, which consists of the 3B segment along with the last seven amino acids of the 3A region. Key amino acids, including Lys-80 and Tyr-89, play crucial roles in stabilizing RNA structures by promoting strand annealing and destabilization (Tang et al, 2014). This RNA chaperone function aids EV-A71 in efficient viral replication by facilitating RNA folding and maintaining replication-competent forms (DeStefano and Titilope, 2006; Tang et al, 2014; Xiang et al, 1995). To further investigate how the chaperone activity of 3AB influences its interaction with mtDNA, we generated truncation and point mutants ($3AB^{\Delta 3B+7}$, $3AB^{K80A}$, and $3AB^{Y89A}$) (Fig. 6A,B). The results of dot blotting and qPCR showed that these mutations significantly reduced the mtDNA-binding affinity of 3AB, indicating the importance of the "3B + 7" region and specific residues for this interaction (Figs. 6C and EV5A). Consistently, these 3AB mutants also failed to bind endogenous mtDNA in TFAM KO cells (Fig. 6D). We then assessed whether these mutations impaired the immunosuppressive function of 3AB. The mutants failed to suppress the poly(dA:dT)-induced IFN-I response, suggesting that the immunosuppressive effect of 3AB is dependent on its ability to bind mtDNA (Figs. 6E and EV5B). Next, we expressed and purified recombinant 3B (Fig. 6F) and investigated whether the 3B protein alone could bind mtDNA. The ELISA results revealed that 3AB, but not 3B, interacted with mtDNA in a dose-dependent manner (Figs. 6G and EV5C). Consistently, dot blotting and qPCR analysis further confirmed that 3AB directly associated with mtDNA (Figs. 6H and EV5D). Collectively, our findings indicate that EV-A71 3AB binds cytosolic mtDNA, thereby interfering with the cGAS–STING pathway and suppressing IFN-I production.

## CVA9 and CVA16 3AB suppress IFN-I signaling via mtDNA binding

To further explore whether this immune evasion mechanism is conserved among other enteroviruses, we examined CVA9. Similar to EV-A71, we observed that CVA9 infection induces mtDNA release (Fig. 7A) and that the CVA9 3AB protein binds mtDNA (Fig. 7B). Additionally, the 3AB protein of CVA16 also exhibited mtDNA-binding capacity (Fig. 7C). To assess the functional consequence of 3AB-mediated mtDNA sequestration on the cGAS–STING pathway, we used TFAM KO cells, which exhibit elevated levels of cytosolic mtDNA. In this system, the 3AB proteins from both CVA9 and CVA16 bound endogenous mtDNA (Fig. 7D) and significantly suppressed IFN-β expression (Fig. 7E), indicating that both proteins

are capable of dampening the mtDNA-mediated cGAS–STING pathway activation. Similarly, both CVA9 and CVA16 3AB proteins inhibited poly(dA:dT)-induced 2'3'-cGAMP production (Fig. 7F). Collectively, these findings suggest that 3AB-mediated interference with cytosolic DNA sensing represents a conserved immune evasion strategy among multiple enteroviruses.

## Discussion

Our study reveals a novel immune evasion mechanism utilized by EV-A71, whereby its nonstructural protein 3AB disrupts the mtDNA–cGAS–STING pathway to evade host innate immunity. The cGAS–STING pathway is a central component of the innate immune response against viral infections triggered by the presence of cytosolic DNA, including mtDNA released from damaged host cells. Recognition of this DNA by cGAS catalyzes the production of the second messenger cGAMP, which in turn activates STING and initiates downstream signaling through TBK1 and IRF3, ultimately driving IFN-I production and an antiviral state (Sun et al, 2013; West et al, 2011; Zhang et al, 2019). Although EV-A71 infection induces mitochondrial damage and the release of mtDNA into the cytoplasm, our findings indicate that EV-A71 effectively blocks the immune response typically elicited by this DAMP.

We found that EV-A71 2B, functioning as a viroporin, induces mtDNA release, consistent with observations for other viral viroporins such as IAV M2 and EMCV 2B. More importantly, we demonstrated that EV-A71 achieves immune evasion through its nonstructural protein 3AB, which directly binds cytosolic mtDNA and prevents its recognition by cGAS. This interaction between 3AB and mtDNA effectively disrupts cGAS–DNA phase separation and blocks the activation of cGAS–STING pathway, thereby inhibiting the downstream production of IFN-I. Immunoprecipitation assays showed a significant reduction in cGAS–mtDNA complex formation in the presence of 3AB, supporting a model in which 3AB competitively binds mtDNA, disrupting cGAS activation. Moreover, the 3AB proteins from CVA9 and CVA16 also exhibited mtDNA-binding ability. These findings identify enteroviral 3AB as a pivotal effector in the immune evasion strategy, facilitating viral persistence within the host.

This strategy is distinct from the immune evasion mechanisms of other viruses that target the cGAS–STING pathway. Some flaviviruses suppress this pathway by degrading or altering the expression of cGAS itself. DENV promotes the lysosomal

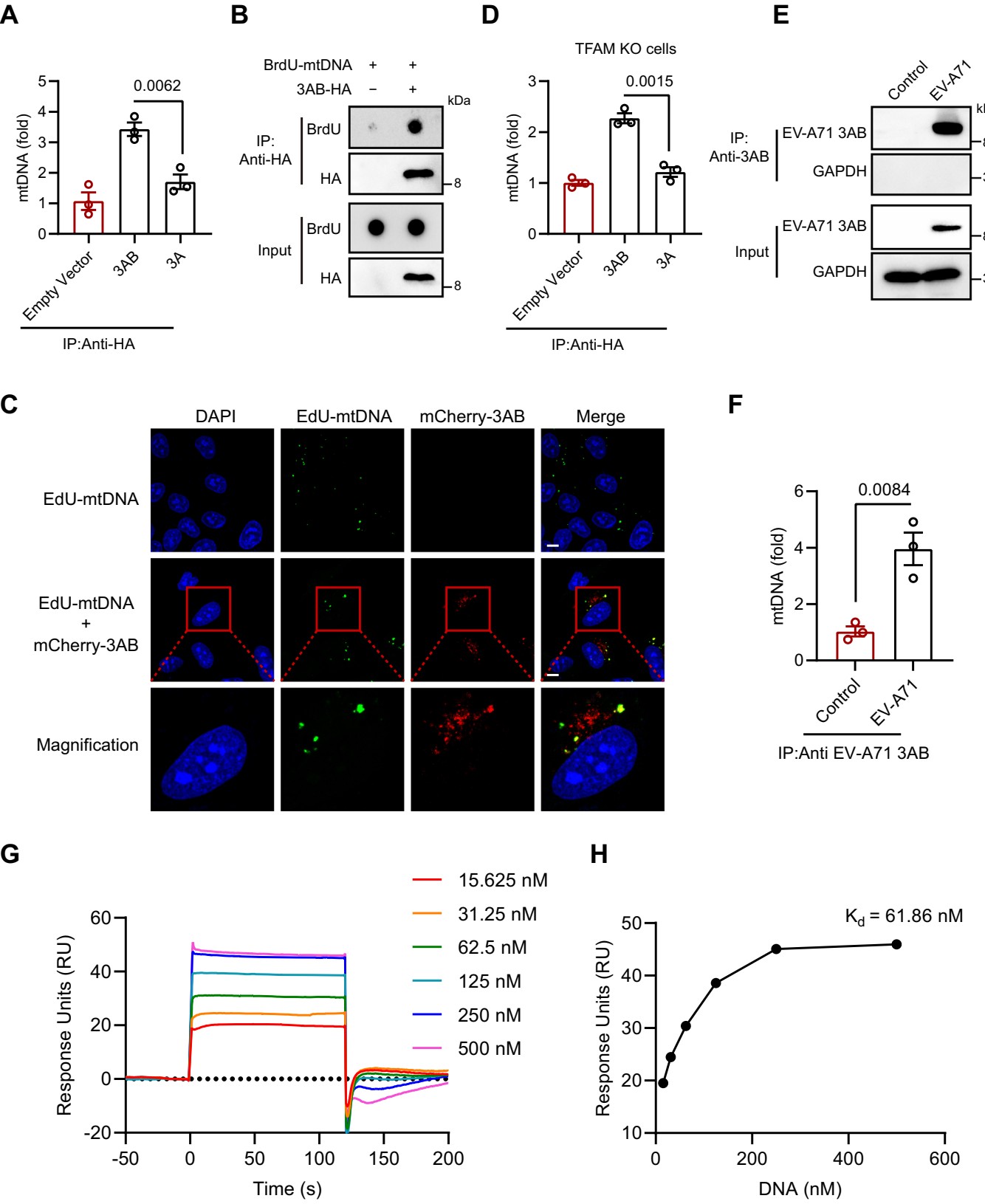

◄ **Figure 5.  EV-A71 3AB protein binds mtDNA.**

(A) After 24 h of co-transfection with 3 AB or 3A plasmid and mtDNA, the cytosol was isolated, followed by immunoprecipitation with anti-HA. qPCR was performed to measure the cytosolic mtDNA levels. Data were presented as mean ± SEM, n = 3 biological replicates. Unpaired t test was used for statistical analysis. (B) After 24 h of co-transfection with 3AB plasmid and mtDNA, the cytosol was isolated, followed by immunoprecipitation with anti-HA. Dot blotting analysis was then used to detect the pulled-down BrdU-mtDNA. (C) Colocalization of mtDNA and 3AB in HeLa cells. Cells were co-transfected with mCherry-3AB plasmid and EdU-mtDNA for 24 h. The nuclei were stained with DAPI. Images were examined with a Nikon C2 confocal microscope. Scale bars, 10 μm. (D) The 3 AB or 3A plasmid was transfected into TFAM knockout and mock-treated cells for 24 h, the cytosol was isolated, followed by immunoprecipitation with anti-HA. qPCR was performed to measure the cytosolic mtDNA levels. Data were presented as mean ± SEM, n = 3 biological replicates. Unpaired t test was used for statistical analysis. (E, F) After 12 h of EV-A71 infection (MOI = 1), the cytosol was isolated, followed by immunoprecipitation with anti-EV-A71 3AB. The immunoprecipitated complexes were analyzed by Western blotting with anti-3AB antibodies (E), and cytosolic mtDNA levels were quantified by qPCR (F). Data were presented as mean ± SEM, n = 3 biological replicates. Unpaired t test was used for statistical analysis. (G, H) SPR analysis of the direct interaction between DNA and 3AB using a Biacore T200 instrument. Recombinant 3AB was immobilized on the SPR sensor chip, and 25-bp dsDNA at different concentrations was injected over the chip. The experiments were repeated at least three times with the similar results. Source data are available online for this figure.

degradation of cGAS, whereas ZIKV recruits USP8 to cleave cGAS, thus blocking the initiation of the immune response (Aguirre et al, 2017; Zheng et al, 2018). In contrast, EV-A71 does not modify cGAS expression or induce its degradation. Instead, by using 3AB to bind and sequester mtDNA, EV-A71 prevents the initial step of immune recognition, representing a unique and indirect approach to cGAS–STING inhibition that may provide an evolutionary advantage by allowing the virus to avoid direct targeting of host immune sensors. KSHV ORF52, HSV-1 VP22, and the SARS-CoV-2 nucleocapsid protein, inhibit cGAS signaling by competitively binding DNA and disrupting cGAS–DNA condensates (Gutmann et al, 2025; Xu et al, 2021). Additionally, previous studies have shown that the EV-A71 2 C protein inhibits STING activation by disrupting its interaction with TBK1 (Liu et al, 2023), further highlighting that EV-A71 employs multiple, complementary strategies to suppress host innate immune responses. In addition to its role in immune suppression, 3AB appears to play a crucial role in promoting EV-A71 replication. As a nucleic acid chaperone, 3AB stabilizes RNA by facilitating proper folding and structural integrity, which may further support efficient viral replication (Tang et al, 2014). This dual function of 3AB in immune evasion and RNA stabilization likely enhances the ability of EV-A71 to persist and replicate within host cells, contributing to the overall pathogenicity of the virus.

In addition to overcoming the cGAS–STING pathway, viruses have evolved other mechanisms to bypass immune barriers, enabling effective replication and spread within the host. Membrane-bound Toll-like receptors (TLRs) and cytoplasmic retinoic acid-inducible gene I (RIG-I)-like receptors (RLRs) are crucial pattern recognition receptors (PRRs) that detect pathogen-associated molecular patterns (PAMPs), including viral RNA genomes released in host cells and double-stranded RNA (dsRNA) generated during viral replication (Goubau et al, 2013; Kawai and Akira, 2011; Lund et al, 2004). CVA6 nonstructural protein 2C targets MDA5 and RIG-I for lysosomal degradation, thus inhibiting IFN-β production (Wang et al, 2023). Enterovirus proteinase 2A (2Apro) proteolytically targets MDA5 and MAVS, whereas proteinase 3C (3Cpro) targets RIG-I (Feng et al, 2014).

Collectively, our findings reveal a multifaceted role of EV-A71 3AB in modulating host immunity and facilitating viral replication. By directly interacting with mtDNA, 3AB inhibits the activation of the cGAS–STING pathway and suppresses IFN-I production, while also promoting viral RNA stability and replication efficiency. This dual functionality highlights 3AB as a potential target for antiviral

therapies aimed at disrupting both immune evasion and viral propagation. Our study not only advances the understanding of the immune evasion tactics of EV-A71 but also underscores the versatility of viral strategies to counteract host defenses, illuminating new avenues for therapeutic intervention against EV-A71 and potentially related viruses (Fig. 8).

## Methods

### Reagents and tools table

| Reagent/ resource | Reference or source | Identifier or catalog number |
|---|---|---|
| **Cell lines** | | |
| HEK293T cells | ATCC | |
| HeLa cells | ATCC | |
| Rhabdomyosarcoma (RD) cells | ATCC | |
| MG-63 cells | National Collection of Authenticated Cell Culture | |
| THP-1 cells | ATCC | |
| **Recombinant DNA** | | |
| pCMV-EV-A71-2B-HA | This study | |
| pCMV-EV-A71-2C-HA | This study | |
| pcDNA3.1-HA-EV-A71 3 A | This study | |
| pCMV-EV-A71-3AB-HA | This study | |
| pCMV-EV-A71-3C-HA | This study | |
| pcDNA3.1-HA-EV-A71-3D | This study | |
| pCMV-EV-A71 3AB $^{\Delta 3B+7}$-HA | This study | |
| pCMV-EV-A71 3AB $^{K80A}$-HA | This study | |
| pCMV-EV-A71 3AB $^{Y89A}$-HA | This study | |
| pCMV-mcherry-EV-A71-3AB | This study | |
| pGEX-6p-GST-EV-A71-3B | This study | |
| pGEX-6p-GST-EV-A71-3AB | This study | |
| pEGFP-C3-cGAS | This study | |
| pT7-6*His-SUMO-cGAS | MiaoLing Plasmid Platform | |
| psPAX2 | Addgene | 12260 |
| pMD2.G | Addgene | 12259 |
| **Antibodies** | | |
| Anti-HA | AbMART | 26D11 |
| Anti-Myc | Medical & Biological Lab (MBL) | 562-5 |

| Reagent/ resource | Reference or source | Identifier or catalog number |
|---|---|---|
| Anti-Flag | Proteintech Group | 20543-1-AP |
| Anti-GST | Proteintech Group | 10000-0-AP |
| Anti-COXIV | Proteintech Group | 11242-1-AP |
| Anti-TFAM | Proteintech Group | 22586-1-AP |
| Anti-BrdU | Proteintech Group | 66241-1-Ig |
| Anti-GAPDH | Proteintech Group | 60004-1-Ig |
| Anti-IRF3 | Proteintech Group | 11312-1-AP |
| Anti-mCherry | Proteintech Group | 26765-1-AP |
| Anti-GFP | Proteintech Group | 50430-2-AP |
| Anti-Enterovirus 71 3AB | Gene Tex | GTX132344 |
| Anti-cGAS | Cell Signaling Technology (CST) | 15102 |
| Anti-STING | Cell Signaling Technology (CST) | 13647 |
| Anti-p-STING (Ser366) | Cell Signaling Technology (CST) | 50907 |
| Anti-p-IRF3 (Ser396) | Cell Signaling Technology (CST) | 29047 |
| Anti-TBK1/NAK | Cell Signaling Technology (CST) | 3504 |
| Anti-p-TBK1/NAK (Ser172) | Cell Signaling Technology (CST) | 5483 |
| **Oligonucleotides and other sequence-based reagents** | | |
| EV-A71-2B forward | GCGTTTAAACTTAAGCTTATG GGCGTGTCCGATTA | |
| EV-A71-2B reverse | TGGATATCTGCAGAATTCCT GCTTTTGAGCGATA | |
| EV-A71-2C forward | GCGTTTAAACTTAAGCTTATGAG CGCTTCCTGGCT | |
| EV-A71-2C reverse | TGGATATCTGCAGAATTCTTGGAAA AGAGCCTCG | |
| EV-A71-3A forward | CCCAAGCTTATGTACCCATAC GATGTTCCAGATTACGCTGGAGGGGG TCCACCCAAGTTCAGG | |
| EV-A71-3A reverse | GGGGATCCCTACTGGAAC CCTGCAAAGAGCTTGTAGATG | |
| EV-A71-3AB forward | GCGTTTAAACTTAAGCTT ATGGGTCCACCCAAGTT | |
| EV-A71-3AB reverse | TGGATATCTGCAGAATTCCT GTACTGTTGCTGTGC | |
| EV-A71-3C forward | GCGTTTAAACTTAAGCTTAT GGGCCCGAGCCTTGA | |
| EV-A71-3C reverse | TGGATATCTGCAGAATTCTT GTTCACTAGCAAAG | |
| EV-A71-3D forward | CCCAAGCTTATGTACCCATA CGATGTTCCAGATTACGCT GGAGGGGGAGAGATCCAGTGGGTT | |
| EV-A71-3D reverse | GGGGATCCAAATAACTCG AGCCAATTGCGTCTAAG | |
| EV-A71-3AB △3B+7 forward | AGCGTTTAAACTTAAGCTTA TGGGTCCACCCAAGTTC | |
| EV-A71-3AB △3B+7 reverse | ATCGTATGGGTAGGATCC GTAGATGACATACACC | |
| EV-A71-3AB K80A forward | TGGTGTACGTCATCTATG CACTCTTTGCAGGGTTT | |
| EV-A71-3AB K80A reverse | AAACCCTGCAAAGAGTGCA TAGATGACGTACACCA | |
| EV-A71-3AB Y89A forward | CAGGGTTTCAGGGTGCGGC ATCTGGTGCCCCCAAA | |
| EV-A71-3AB Y89A reverse | TTTGGGGGCACCAGATGC CGCACCCTGAAACCCTG | |
| mcherry-3AB forward | TCCGGACTCAGATCTCGA GCTGGTCCACCCAAGTTC | |
| mcherry-3AB reverse | TACCGTCGACTGCAGAAT TCTTACTGTACTGTTGCTG | |
| GST- EV-A71-3B forward | GATCCGGTGCGTATTCTGGT GCTCCTAAGCAAGTGCTTAA GAAACCTGCTCTTCGCAC AGCAACAGTACAGTAAC | |
| GST- EV-A71-3B reverse | TCGAGTTACTGTACTGTTGC TGTGCGAAGAGCAGGTTTC TTAAGCACTTGCTTAGG AGCACCAGAATACGCACCG | |
| 100bp-DNA sense | ACATCTAGTACATGTCTAGTC AGTATCTAGTGATTATCTA GACATACATCTAGTACATG TCTAGTCAGTATCTAGTG ATTATCTAGACATGGACTCATCC | |
| 100bp-DNA antisense | GGATGAGTCCATGTCTAGAT AATCACTAGATACTGACTAG ACATGTACTAGATGTATGTCT AGATAATCACTAGATACTGAC TAGACATGTACTAGATGT | |
| 100 bp Cy3-DNA sense | ACATCTAGTACATGTCTAGT CAGTATCTAGTGATTATCTA GACATACATCTAGTACATGT CTAGTCAGTATCTAGTGATT ATCTAGACATGGACTCATCC | |
| 100 bp Cy3-DNA antisense | GGATGAGTCCATGTCTAGA TAATCACTAGATACTGACT AGACATGTACTAGATGTAT GTCTAGATAATCACTAGAT ACTGACTAGACATGTACTAGATGT | |
| 25bp-DNA sense | AAAACAAACAACACAACAAACAAAA | |
| 25bp-DNA antisense | TTTTGTTTGTTGTGTTGTTTGTTTT | |
| **qPCR Primer** | | |
| Human Actin forward | TGACGTGGACATCCGCAAAG | |
| Human Actin reverse | CTGGAAGGTGGACAGCGAGG | |
| Human IFN-β forward | CTCCTGGCTAATGTCTATCA | |
| Human IFN-β reverse | GCAGAATCCTCCCATAATAT | |
| EV-A71 VP1 forward | GCAGCCCAAAAGAACTTCAC | |
| EV-A71 VP1 reverse | ATTTCAGCAGCTTGGAGTGC | |
| Human mtDNA forward | CACCCAAGAACAGGGTTTGT | |
| Human mtDNA reverse | TGGCCATGGG TATGTTGTTAA | |
| **sgRNA** | | |
| Human TFAM | GCGTTTCTCC GAAGCATGTG | |
| **Chemicals, enzymes and other reagents** | | |
| DMEM | Gibco | 11995500 |
| FBS | ExCell Bio | FSP500 |
| RPMI 1640 | Gibco | 22400089 |
| trypsin | Thermo Fisher | 25200 |
| Site-Directed Mutagenesis Kit | Beyotime | D0206S |
| Lipofectamine 2000 | Invitrogen | 11668-019 |
| JC-10 mitochondrial membrane potential assay kit | Yeasen | 40752ES60 |
| Digitonin | Beyotime | ST1272 |
| DNA Isolation Mini Kit | Vazyme | DC112 |
| BrdU | Beyotime | ST1056 |
| EdU | Beyotime | C0071S |
| Cell Mitochondria Isolation Kit | Beyotime | C3601 |
| Dual Luciferase Reporter Gene Assay Kit | Beyotime | RG027 |
| Pierce Classic IP Kit | Thermo Scientific | 26146 |
| Protein A/G agarose | Beyotime | P2055 |

| Reagent/ resource | Reference or source | Identifier or catalog number |
|---|---|---|
| PMSF | Beyotime | ST506 |
| RNeasy Mini Kit | Axygen | 17921KD1 |
| HisScript III RT SuperMix for qPCR | Vazyme | R323-01 |
| qPCR SYBR | Yeasen | 11201ES08 |
| M-PER Reagent | Thermo Scientific | 78501 |
| 2'3'-cGAMP ELISA kit | Cayman | 501700 |
| Human IFN-β ELISA kit | Solarbio | SEKH-0410 |
| Human CXCL10/IP10 ELISA kit | Solarbio | SEKH-0070 |
| 4% Paraformaldehyde Fix Solution | Beyotime | P0099 |
| Triton X-100 | Biosharp | BS084 |
| DAPI | Beyotime | C1005 |
| TMB Microwell Peroxidase Substrate System | SeraCare | 5120-0053 |
| GST spin purification kit | Beyotime | P2262 |
| His-tag Protein Purification Kit | Beyotime | P2226 |
| CM5 sensory chips | cytiva | 29104988 |
| PBS-P+ buffer | cytiva | 28995084 |
| FITC Quick Labeling Kit | Beyotime | P0639S |
| AF647 Quick Labeling Kit | Beyotime | P1265S |
| ATP | Beyotime | D7378 |
| GTP | Beyotime | D7380 |
| BL21 | Solarbio | C1400 |
| BSA | Solarbio | 9048-46-8 |
| MgCl₂ | Macklin | 7786-30-3 |
| IPTG | Beyotime | ST098 |
| RU.521 | MCE | HY-114180 |
| H151 | MCE | HY-112693 |
| diABZI | sellcek | S8796 |
| HT-DNA | Merck/Sigma-Aldrich | D6898 |
| Recombinant cGAS | MCE | HY-P72337 |
| **Software** | | |
| GraphPad Prism 9 | https://www.graphpad.com/ | |
| Snapgene 6.0.2 | https://www.snapgene.com/ | |
| LAS-X 4.6.0 | Leica Microsystems | |
| BIA evaluation software | https://www.cytivalifesciences.com/ | |
| ImageJ | Imagej.nih.gov/index/html | |
| Image Lab (Bio-Rad) | https://www.bio-rad.com/ | |
| **Other** | | |
| Leica STELLARIS 5 confocal microscope | Leica | |
| Nikon C2 confocal microscopy | Nikon | |
| Varioskan LUX | Thermo Scientific | |
| CFX Opus 96 | Bio-Rad | |
| Biacore T200 | Cytiva | |
| CHEMIDOCIMAGINGSYSTEM | Bio-Rad | |
| GelDox XR+ | Bio-Rad | |

## Cells and virus

HEK293T cells, HeLa cells, rhabdomyosarcoma (RD) cells and MG-63 cells were cultured in Dulbecco's modified Eagle's medium (DMEM) (Cat# 11995500, Gibco) supplemented with 10% heat-inactivated fetal bovine serum (FBS) (Cat# FSP500, ExCell Bio) at 37 °C with 5% $CO_2$. THP-1 cells were maintained in Roswell Park Memorial Institute (RPMI) 1640 medium (Cat# 22400089, Gibco) supplemented with 10% heat-inactivated fetal bovine serum. Cells are regularly tested for mycoplasma. EV-A71 and CVA9 were stored in our laboratory. The viruses were passaged in RD cells. The virus titers were determined by a 50% tissue culture infectious dose ($TCID_{50}$) assay.

## Western blotting, dot blotting analysis and antibodies

The samples were separated by 12% SDS–PAGE, electrophoretically transferred to polyvinylidene fluoride membranes, which were blocked and incubated with primary antibodies. For dot blotting, the samples were spotted onto nitrocellulose membranes and cross-linked with ultraviolet light, followed by the same steps as those used for Western blotting. The following antibodies were used in these experiments: an anti-HA antibody (Cat# 26D11) was obtained from AbMART; an anti-Myc antibody (Cat# 562-5) was purchased from Medical & Biological Lab (MBL); an anti-Enterovirus 71 3AB (Cat# GTX132344) was obtained from Gene Tex; anti-Flag (Cat# 20543-1-AP), anti-COXIV (Cat# 11242-1-AP), anti-TFAM (Cat# 22586-1-AP), anti-BrdU (Cat# 66241-1-Ig), anti-GAPDH (Cat# 60004-1-Ig), anti-GST (Cat# 10000-0-AP), anti-IRF3 (Cat# 11312-1-AP), anti-mCherry (Cat# 26765-1-AP) and anti-GFP (Cat# 50430-2-AP) antibodies were obtained from the Proteintech Group; and anti-cGAS (Cat# 15102), anti-STING (Cat# 13647), anti-p-STING (Ser366) (Cat# 50907), anti-p-IRF3 (Ser396) (Cat# 29047), anti-TBK1/NAK (Cat# 3504), and anti-p-TBK1/NAK (Ser172) (Cat# 5483) antibodies were obtained from Cell Signaling Technology (CST).

## Plasmids

The full-length viral cDNA of EV-A71 was inserted into the pCMV-HA or pcDNA3.1(+) vector to construct plasmids expressing HA-tagged viral nonstructural proteins. The EV-A71 3AB gene was inserted into the pCMV-mCherry vector for cellular localization experiments. Mutant plasmids containing the EV-A71 3AB protein were constructed according to the manufacturer's instructions of the site-directed gene mutation kit (Cat# D0206S, Beyotime). The genes encoding CVA9 3AB and CVA16 3AB were chemically synthesized by Genscript Biotech Corporation. All the constructed plasmids were analyzed and verified by DNA sequencing. The plasmids were transfected into the cells with Lipofectamine 2000 transfection reagent (Cat# 11668-019, Invitrogen) according to the manufacturer's protocol.

## Mitochondrial membrane potential

The mitochondrial membrane potential was assessed with a JC-10 mitochondrial membrane potential assay kit (Cat# 40752ES60, Yeasen) according to the manufacturer's instructions. Briefly, the cells were mock-infected or infected with EV-A71 and collected and stained with JC-10 (1:200) at 37 °C for 30 min. In healthy mitochondria, JC-10 accumulates in the mitochondrial matrix and forms aggregates that emit red fluorescence (Ex/Em = 540/590 nm). In contrast, in depolarized or damaged mitochondria, JC-10

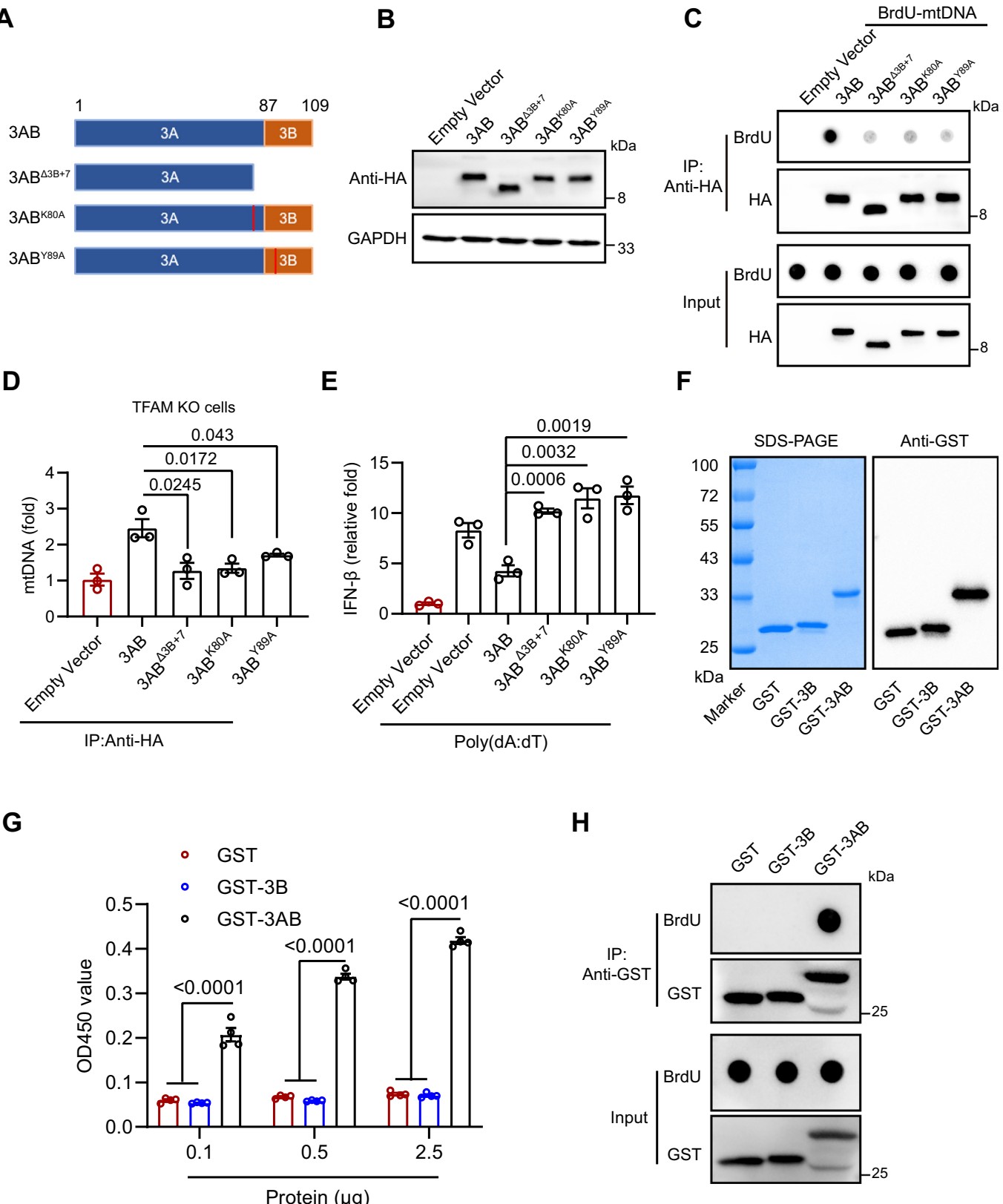

◀ **Figure 6. Mapping the functional domains of EV-A71 3AB protein responsible for mtDNA binding.**

(A, B) Schematic representation of 3AB mutants. Three 3AB mutants plasmids were constructed and expressed in HeLa cells. (C) After 24 h of co-transfection with 3AB mutants plasmids and mtDNA, the cytosol was isolated, followed by immunoprecipitation with anti-HA. Dot blotting analysis was then used to detect the pulled-down BrdU-mtDNA. (D) The 3AB mutants plasmids were transfected into TFAM knockout and mock-treated cells for 24 h, the cytosol was isolated, followed by immunoprecipitation with anti-HA. qPCR was performed to measure the cytosolic mtDNA levels. Data were presented as mean ± SEM, $n = 3$ biological replicates. Unpaired $t$ test was used for statistical analysis. (E) HeLa cells were transfected with 100 ng/mL poly(dA:dT) for 6 h, followed by transfection with 3AB mutants plasmids for 24 h. qRT–PCR was performed to measure IFN-β mRNA expression. Data were presented as mean ± SEM, $n = 3$ biological replicates. Unpaired $t$ test was used for statistical analysis. (F) Purification of the recombinant GST, GST-3B, and GST-3AB proteins produced in the *Escherichia coli* BL21 DE3 strain. The proteins were detected by Coomassie blue staining in SDS–PAGE gels and further confirmed by Western blotting analysis. (G) A direct binding ELISA assay was performed to detect the interaction between 3AB protein and mtDNA. Purified GST-tagged proteins (GST, GST-3B, GST-3AB) were coated onto a 96-well ELISA plate and incubated overnight at 4 °C. BrdU-labeled mtDNA was then added and incubated for 2 h at room temperature. After washing, the bound mtDNA was detected using anti-BrdU antibodies. Data were presented as mean ± SEM, $n = 4$ biological replicates. Unpaired $t$ test was used for statistical analysis. (H) Purified GST-3AB protein and mtDNA were incubated together in binding buffer (40 mM tris-acetate (pH 7.5), 20 mM KCL, 10 mM MgCl$_2$, 10% glycerol). The protein–mtDNA complexes were immunoprecipitated using anti-GST antibodies. Dot blotting analysis was then performed to assess the pulled-down BrdU-mtDNA. The experiments were repeated at least three times with the similar results. Source data are available online for this figure.

remains in its monomeric form in the cytoplasm, emitting green fluorescence (Ex/Em = 490/525 nm). The ratio of red to green fluorescence serves as an indicator of mitochondrial membrane potential, with a decreased ratio reflecting mitochondrial depolarization. The red/green fluorescence ratio was quantified using a fluorescence microplate reader, and images were captured using a Nikon C2 confocal microscopy or Leica STELLARIS 5 confocal microscope.

## Detection of mtDNA in cytosolic extracts

RD, U251, THP-1, and HeLa cells were resuspended in 500 μL of buffer (150 mM NaCl, 50 mM HEPES (pH 7.4), and 20 μg/mL digitonin). The homogenates were incubated on an end-over-end rotator for 10 min. After centrifugation (1000×$g$, 3 min) three times, the supernatants were transferred to fresh tubes and centrifuged at 16,000×$g$ for 20 min. Cytosolic mtDNA was isolated from these pure cytosolic fractions with a DNA Isolation Mini Kit (Cat# DC112, Vazyme). Total mtDNA was isolated from whole-cell extracts with a DNA Isolation Mini Kit (Cat# DC112, Vazyme). Quantitative PCR (qPCR) was performed on cytosolic fractions and whole-cell extracts using mtDNA primers targeting the MT-TL1 gene, which encodes the mitochondrial transfer RNA for leucine 1 (UUA/G). The relative cytosolic mtDNA levels were normalized to total mtDNA amounts.

## Isolation of BrdU- and EdU-labeled mitochondrial DNA

HeLa cells were treated with BrdU (Cat# ST1056, Beyotime) or EdU (Cat# C0071S, Beyotime) at a concentration of 10 μM for 48 h. The labeled cells were washed three times with PBS, detached from the culture flask with trypsin/EDTA solution (Cat# 25200, Thermo Fisher), and centrifuged at 600×$g$ for 5 min. The supernatant was then removed, and the cells were resuspended in cell lysis reagent (Cat# C3601-1, Beyotime) and incubated in an ice bath for 15 min. The lysed cells were homogenized with a glass homogenizer for 30 cycles and then centrifuged at 1000×$g$ for 10 min at 4 °C. The supernatant was collected and resuspended in cell lysis buffer before being centrifuged again at 1000×$g$ for 10 min at 4 °C for additional purification. Finally, the supernatant was collected and centrifuged at 3500×$g$ for 10 min at 4 °C. The resulting pellet contained the isolated mitochondria. Mitochondrial storage reagent (Cat# C3601-3, Beyotime) was used

to resuspend the isolated mitochondria at the required concentration for further experiments.

## Luciferase activity assays

For luciferase activity assays, HEK-293T cells were co-transfected with IFN- β-Luc, pRL-TK, Myc-cGAS, STING-Flag, and 3AB-HA plasmids with Lipofectamine 2000 transfection reagent (Cat# 11668-019, Invitrogen) for 24 h. Cells were harvested for luciferase activity analysis with the Dual-Luciferase Reporter Assay System (Cat# RG027, Beyotime). Luminescence was measured with a microplate reader (Thermo Scientific). The firefly luciferase activity was normalized to the control Renilla luciferase activity.

## Coimmunoprecipitation (co-IP)

Recombinant plasmids were transiently transfected into 293 T cells with Lipofectamine 2000 transfection reagent (Cat# 11668-019, Invitrogen). After 24 h, coimmunoprecipitation (co-IP) was carried out with a Pierce Classic IP Kit (Cat# 26146, Thermo Scientific). The cells were lysed on ice with IP lysis buffer containing a protease inhibitor cocktail (Cat# ST506, Beyotime). The cell lysates were incubated overnight with 2 μg of antibody. Protein A/G agarose (Cat# P2055, Beyotime) was then added for 3 h of incubation. Following washing with IP lysis buffer, protein samples were analyzed by Western blotting, and mtDNA was detected by dot blotting. The pulled-down mtDNA was isolated from the protein complexes with the DNA Isolation Mini Kit (Cat# DC112, Vazyme). Quantitative PCR was performed with mtDNA-specific primers.

## Establishment of knockout cell lines with the CRISPR/Cas9 system

Small guide RNAs (sgRNAs) targeting human TFAM, were ligated into the Lenti-CRISPRv2 plasmid. The Lenti-CRISPRv2 plasmid (with sgRNA cloned) was co-transfected into HEK-293T cells with the packaging plasmids psPAX2 and pMD2.G for 48–72 h to generate lentivirus. HeLa cells were infected with the recombinant lentivirus for 48 h, followed by selection with puromycin (1 μg/mL) for 3–5 days. The sgRNA sequences are listed in Reagents and Tools Table.

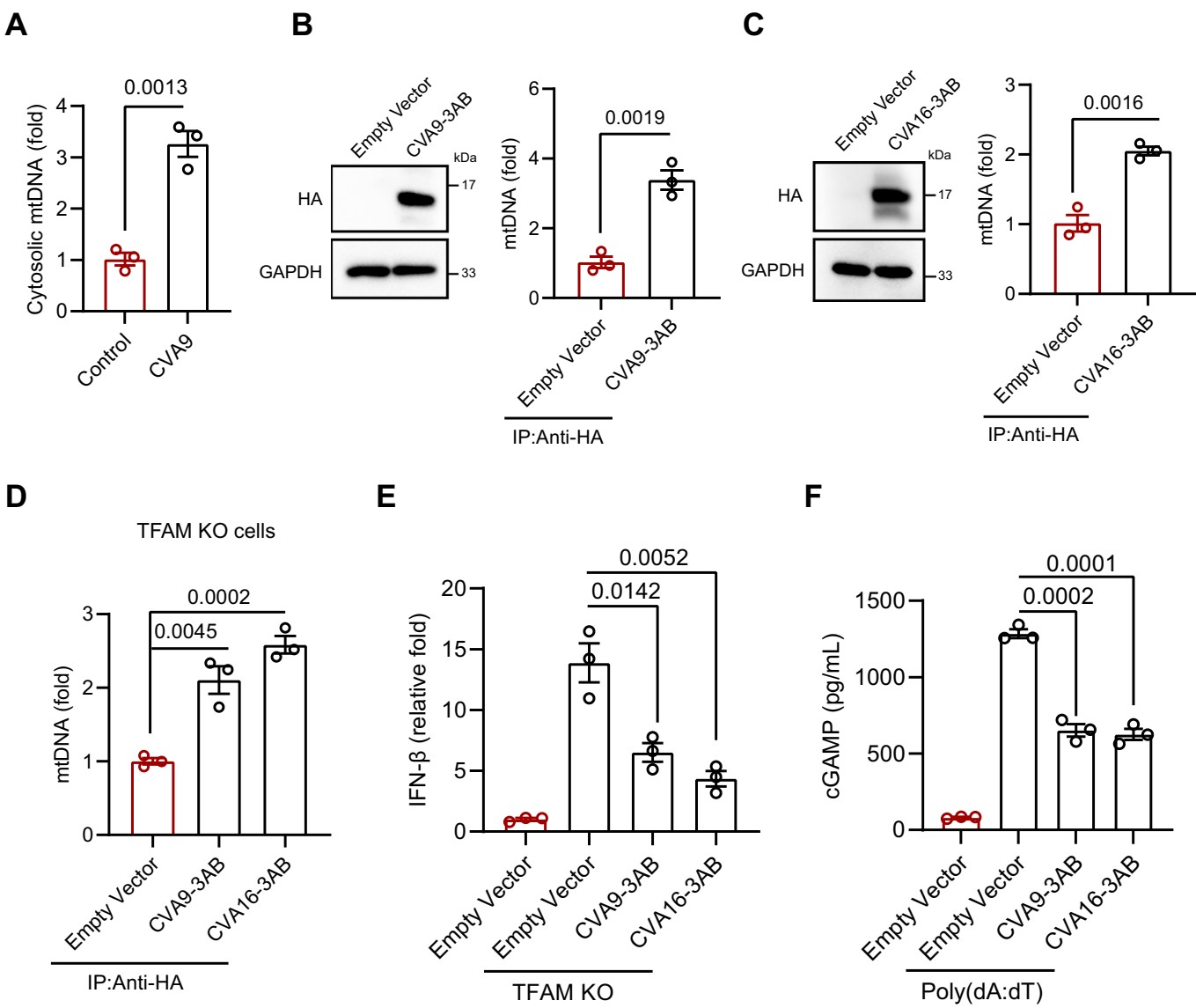

**Figure 7. CVA9 and CVA16 3AB inhibit IFN-I signaling by targeting cytosolic mtDNA.**

(A) HeLa cells were infected with CVA9 (MOI = 1) for 24 h. Cytosolic fractions were isolated, and qPCR was conducted to measure the cytosolic mtDNA. Data were presented as mean ± SEM, $n = 3$ biological replicates. Unpaired $t$ test was used for statistical analysis. (B, C) After 24 h of co-transfection with CVA9 or CVA16 3AB plasmid and mtDNA, the cytosol was isolated, followed by immunoprecipitation with anti-HA. qPCR was performed to measure the cytosolic mtDNA levels. Data were presented as mean ± SEM, $n = 3$ biological replicates. Unpaired $t$ test was used for statistical analysis. (D) The CVA9 3AB or CVA16 3AB plasmid was transfected into TFAM knockout and mock-treated cells for 24 h, the cytosol was isolated, followed by immunoprecipitation with anti-HA. qPCR was performed to measure the cytosolic mtDNA levels. Data were presented as mean ± SEM, $n = 3$ biological replicates. Unpaired $t$ test was used for statistical analysis. (E) The CVA9 3AB or CVA16 3AB plasmid was transfected into TFAM knockout and mock-treated cells for 24 h, after which IFN-β mRNA expression was measured by qRT-PCR. Data were presented as mean ± SEM, $n = 3$ biological replicates. Unpaired $t$ test was used for statistical analysis. (F) HeLa cells were transfected with 100 ng/mL poly(dA:dT) for 6 h, followed by transfection with CVA9 3AB or CVA16 3AB plasmid for 24 h. Intracellular cGAMP levels were measured by ELISA. Data were presented as mean ± SEM, $n = 3$ biological replicates. Unpaired $t$ test was used for statistical analysis. The experiments were repeated at least three times with the similar results. Source data are available online for this figure.

## RNA extraction and quantitative PCR (qPCR)

Total RNA was isolated from cells with an RNeasy Mini Kit (Cat# 17921KD1, Axygen) and reverse-transcribed into cDNA with an iScript cDNA synthesis kit (Cat# R323-01, Vazyme). The quantity of RNA was normalised by Actin as reference gene. qRT-PCR was performed on a Bio-Rad CFX96 Touch Real-Time Detection System. The qRT-PCR primers used are listed in Reagents and Tools Table.

## Quantification of the cGAMP concentration

The appropriate amount of M-PER Reagent (Cat# 78501, Thermo Scientific) was added to EV-A71 infected cells, which were then

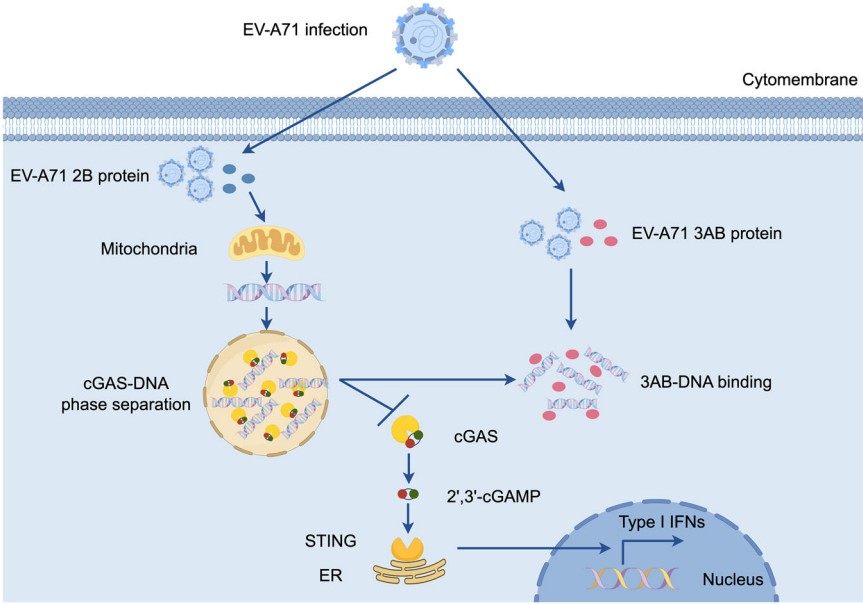

**Figure 8. Schematic model of EV-A71 3AB protein blocking cGAS recognition of mtDNA.**

During EV-A71 infection, the viral 2B protein promotes the release of mtDNA into the cytoplasm, while the 3AB protein competitively binds to mtDNA, thereby preventing cGAS recognition. This blockade disrupts cGAS–DNA phase separation and suppresses cGAS–STING-mediated IFN-I responses.

centrifuged at ~14,000×g for 5–10 min to pellet the cell debris. The supernatant was transferred to a new tube for analysis. The level of 2'3'-cGAMP was detected with a 2'3'-cGAMP ELISA kit (Cayman Chemical Co., Ann Arbor, MI), and the optical density at 450 nm was measured with an ELISA reader.

## Immunofluorescence staining and microscopy

The cells were washed three times in phosphate-buffered saline (PBS) and fixed with 4% paraformaldehyde (Cat# P0099, Beyotime) for 15 min following the antibody instructions. The cells were then permeabilized with 0.3% Triton X-100 (Cat# BS084, Biosharp) for 20 min, washed with PBS, and blocked in 2% BSA for 1 h. The nuclei were stained blue with DAPI (Cat# C1005, Beyotime). Images were examined with a Leica STELLARIS 5 confocal microscope.

## Enzyme-linked immunosorbent assay (ELISA)

The purified proteins and mtDNA were subjected to an enzyme-linked immunosorbent assay (ELISA). The prey protein (2 µg) was coated and incubated overnight at 4 °C. The plates were blocked with 5% w/v bovine serum albumin (BSA) solution for 1 h at room temperature. Bait mtDNA or protein was added, and the mixture was incubated for 2 h. After washing, the primary antibody was added, and the mixture was incubated for an additional 2 h. After washing again, secondary IgG-HRP was added to each well, and the plates were incubated for 1 h. A commercial peroxidase substrate system was used for signal detection (Cat# 5120-0053, SeraCare), and the optical density at 450 nm was measured with an ELISA reader.

For cytokine measurements, cell culture supernatants were collected and the concentrations of IFN-β and CXCL10 were quantified using human IFN-β and CXCL10 ELISA kits (Cat# SEKH-0410, Cat# SEKH-0070, Solarbio), respectively, according to the manufacturer's instructions.

## Expression and purification of recombinant protein

Gene encoding EV-A71 3AB was chemically synthesized by Genscript Biotech Corporation. The 3B gene was cloned and inserted into a pGEX-6p-GST expression vector, with the cloning primers provided in Reagents and Tools Table. pT7-6×His-SUMO-cGAS plasmid (Cat# P44631) was obtained from the MiaoLing Plasmid Platform. Recombinant 3AB and 3B proteins were expressed in the *Escherichia coli* BL21 DE3 strain and purified with a GST spin purification kit (Cat# P2262, Beyotime), while recombinant cGAS protein was expressed in the same strain and purified using a His-tag protein purification kit (Cat# P2226, Beyotime), according to the manufacturer's instructions. The purified proteins were then stained with Coomassie Brilliant Blue. Recombinant cGAS and 3AB proteins were labeled with FITC and Alexa Fluor 647 (Beyotime), respectively, and used in subsequent in vitro phase separation assays and imaging.

## Surface plasmon resonance (SPR) analysis

The binding affinity between 3AB protein and DNA was measured using a Biacore T200 instrument (Cytiva). Recombinant 3AB protein was covalently coupled to the CM5 sensory chips via amine coupling to a density of approximately 1000 response units (RU). Binding measurements were performed at 25 °C and a flow rate of 30 µL/min. DNA (25-bp) was diluted in PBS containing 0.05% Surfactant P20 (Cat# 28995084, Cytiva) supplemented with 10 mM MgCl$_2$ and injected over the chip for 120 s, followed by a 180 s

dissociation phase. The 25-bp DNA used are listed in Reagents and Tools Table.

## In vitro phase separation assay

Recombinant cGAS protein (20 µM, 3% FITC-labeled) was mixed with 100-bp DNA (10 µM, 2% Cy3-labeled) in tubes pre-coated with 20 mg/mL BSA (Solarbio). Mixtures were incubated in buffer containing 20 mM Tris-HCl (pH 7.5), 150 mM NaCl, 10 mM $MgCl_2$ and 1 mg/mL BSA at room temperature. Phase separation was imaged using a Leica STELLARIS 5 confocal microscope. The 100-bp DNA used are listed in Reagents and Tools Table.

## In vitro disruption of cGAS–DNA phase separation

Recombinant cGAS (20 µM, 3% FITC-labeled) and 100-bp dsDNA (10 µM, 2% Cy3-labeled) were first incubated for 10 min to form phase separation, followed by the addition of 20 µM viral protein EV-A71 3AB (3% Alexa Fluor 647-labeled). Time-lapse imaging was performed using a Leica STELLARIS 5 confocal microscope, with images acquired every 15 s for 15 min. The 100-bp DNA used are listed in Reagents and Tools Table.

## In vitro cGAS activity assay

Recombinant cGAS (0.5 µg/mL, Cat# HY-P72337, MCE) was incubated in a reaction mixture containing 20 mM Tris-HCl (pH 7.4), 5 mM $MgCl_2$, 0.2 mg/mL BSA, 0.5 ng/µL 100-bp DNA, 1 mM ATP, and 1 mM GTP. After 1 h of incubation at 37 °C, the samples were boiled for 10 min and centrifuged at 17,000×$g$ for 5 min. The supernatant was collected to measure cGAMP concentrations using a cGAMP ELISA kit (Cayman Chemical Co., Ann Arbor, MI). The 100-bp DNA used are listed in Reagents and Tools Table.

## Quantification and statistical analysis

GraphPad Prism 9 was employed for statistical analysis. The data were presented as the mean ± SEM and were statistically analyzed with a two-tailed unpaired Student's $t$ test. A $P$ value of <0.05 was considered to indicate statistical significance.

# Data availability

All data generated or analyzed during this study are included in this published article and its source data files.

The source data of this paper are collected in the following database record: biostudies:S-SCDT-10_1038-S44319-026-00756-x.

# Peer review information

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

## Acknowledgements

This work was funded by the grants from the National Key Research and Development Plan of China (2023YFD1801900), the National Natural Science Foundation of China (82102375, 32470175, 31972680), Zhejiang Provincial Natural Science Foundation (LQ22 C010007, ZCLQN25H1901), Ningbo Natural Science Foundation (2024J466), the Summit Advancement Disciplines of Zhejiang Province (Wenzhou Medical University–Pharmaceutics), and Science Technology Department of Wenzhou (2023S0094). We thank the core facilities of the Scientific Research Center for technical assistance (Wenzhou Medical University).

## Author contributions

**Peng Sun**: Conceptualization; Funding acquisition; Visualization; Writing—original draft; Writing—review and editing. **Xinya Yang**: Data curation; Formal analysis; Investigation. **Jing Cui**: Data curation; Investigation. **Guicun Fang**: Investigation. **Yuqin Wu**: Investigation. **Jingyi Chang**: Investigation. **Xiaofei Li**: Investigation. **Yinli Xie**: Writing—review and editing. **Lipeng Gan**: Writing—review and editing. **Lina Ma**: Writing—review and editing. **Zhiyong Li**: Conceptualization; Funding acquisition; Writing—original draft; Writing—review and editing.

Source data underlying figure panels in this paper may have individual authorship assigned. Where available, figure panel/source data authorship is listed in the following database record: biostudies:S-SCDT-10_1038-S44319-026-00756-x.

## Disclosure and competing interests statement

The authors declare no competing interests.

# Expanded View Figures

**Figure EV1.   EV-A71 2B protein induces mitochondrial damage.**

(**A–D**) HeLa, U251, RD, and THP-1 cells were infected with EV-A71 (MOI = 1) for 12, 12, 6, and 24 h, respectively. Cytosolic fractions were isolated, and whole-cell lysates (WCL) or cytosolic fractions (Cyt) were analyzed by Western blotting. (**E**) HeLa cells were infected with EVA-71-UV for 12 h. Cytosolic fractions were isolated, and qPCR was conducted to measure the cytosolic mtDNA. Data were presented as mean ± SEM, $n = 3$ biological replicates. Unpaired $t$ test was used for statistical analysis. (**F**) Mitochondrial morphology analyzed in RD cells transfected with EV-A71 2B plasmid. After 18 h of EV-A71 2B transfection in RD cells, mitochondrial morphology was examined by transmission electron microscopy. Scale bars, 1 µm. (**G**) After 18 h of EV-A71 2B plasmid transfection in HeLa or RD cells, the mitochondrial membrane potential was assessed with JC-10, with CCCP serving as a positive control. Images were examined with a Leica STELLARIS 5 confocal microscope (left panel). Scale bars, 10 µm. The ratio of red to green fluorescence for JC-10 was analyzed with a multifunctional microplate reader (right panel). Data were presented as mean ± SEM, $n = 3$ biological replicates. Unpaired $t$ test was used for statistical analysis. (**H**) HeLa cells were transfected with the EV-A71 2B plasmid for 24 h. Western blotting analysis was then conducted to assess the levels of phosphorylated (p-) and total TBK1, as well as total cGAS, STING and IRF3. The experiments were repeated at least three times with the similar results. Source data are available online for this figure.

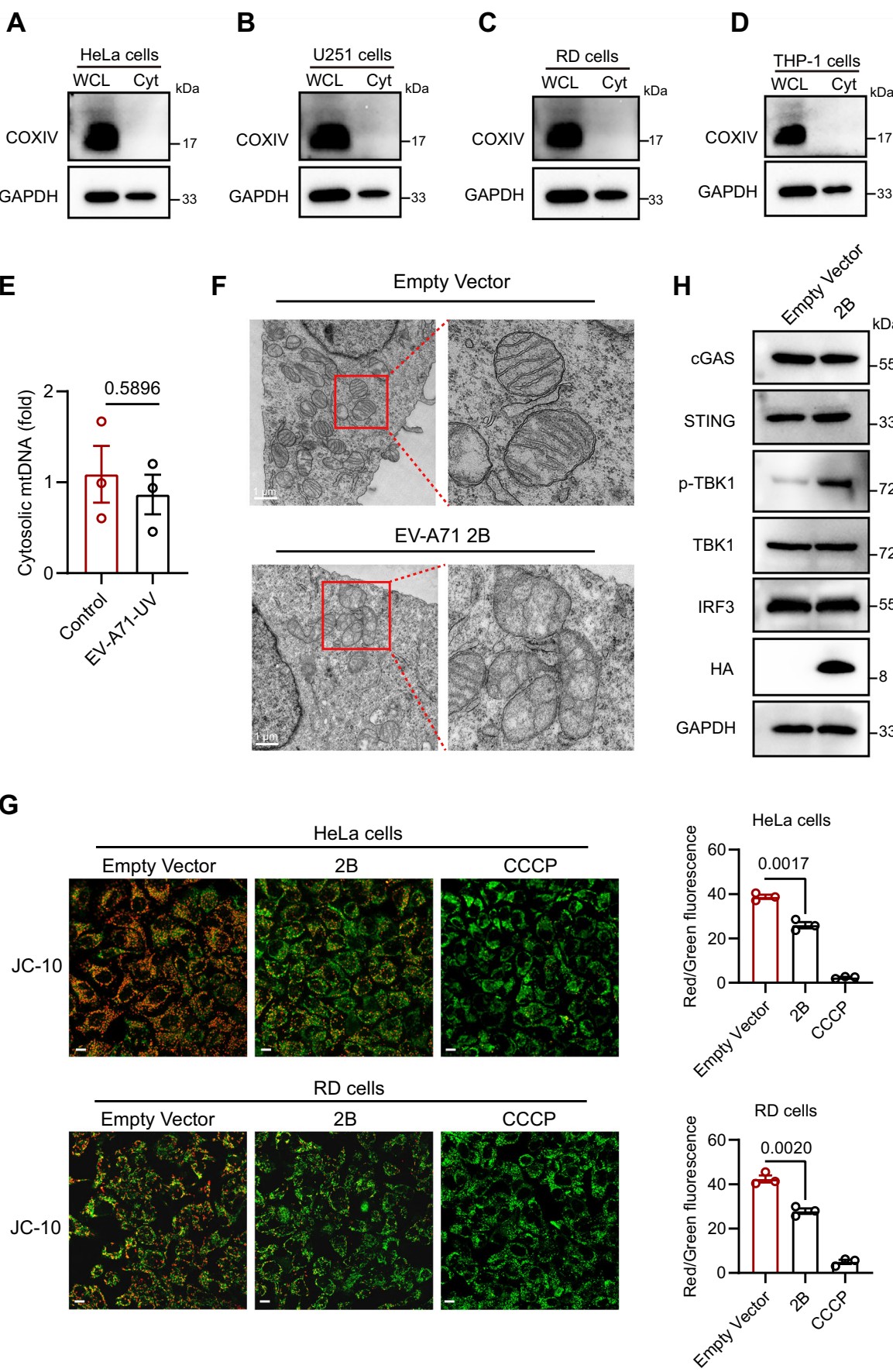

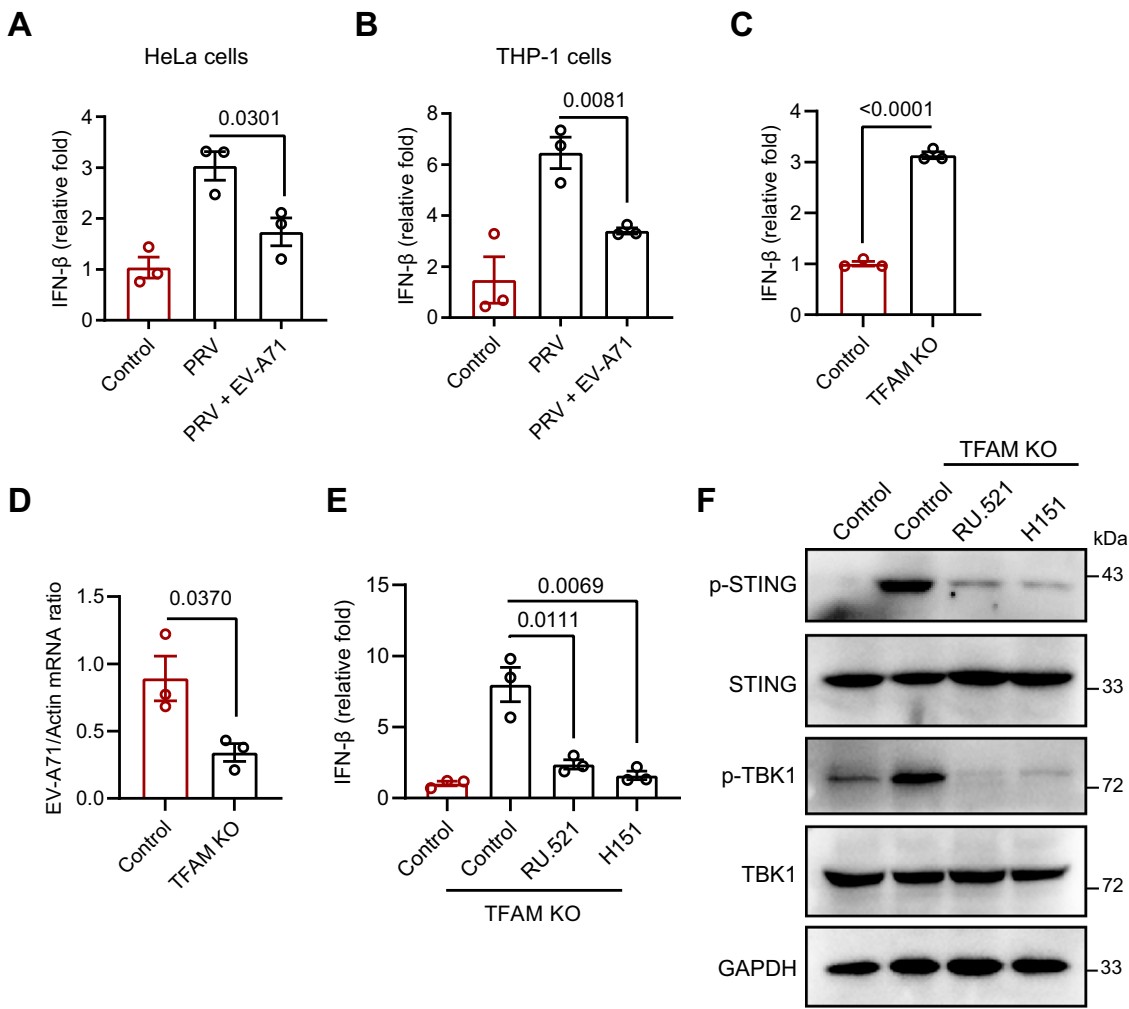

**Figure EV2. TFAM knockout enhances IFN-I expression and suppresses EV-A71 infection.**

(A, B) HeLa and THP-1 cells were infected with PRV for 12 h, followed by EV-A71 infection (MOI = 1) for 12 h and 24 h, respectively. qRT–PCR was then performed to measure the expression of IFN-β mRNA. Data were presented as mean ± SEM, n = 3 biological replicates. Unpaired t test was used for statistical analysis. (C) IFN-β mRNA expression was detected by qRT–PCR in TFAM knockout cells. Data were presented as mean ± SEM, n = 3 biological replicates. Unpaired t test was used for statistical analysis. (D) Intracellular viral load was assessed at 12 h post-infection using qRT–PCR in TFAM knockout cells. Data were presented as mean ± SEM, n = 3 biological replicates. Unpaired t test was used for statistical analysis. (E, F) TFAM knockout cells were treated with RU.521 (10 μM) or H151 (1 μM) for 4 h, after which IFN-β mRNA expression was measured by qRT–PCR (E), and protein levels of total and phosphorylated TBK1 and STING were analyzed by western blotting (F). Data were presented as mean ± SEM, n = 3 biological replicates. Unpaired t test was used for statistical analysis. The experiments were repeated at least three times with the similar results. Source data are available online for this figure.

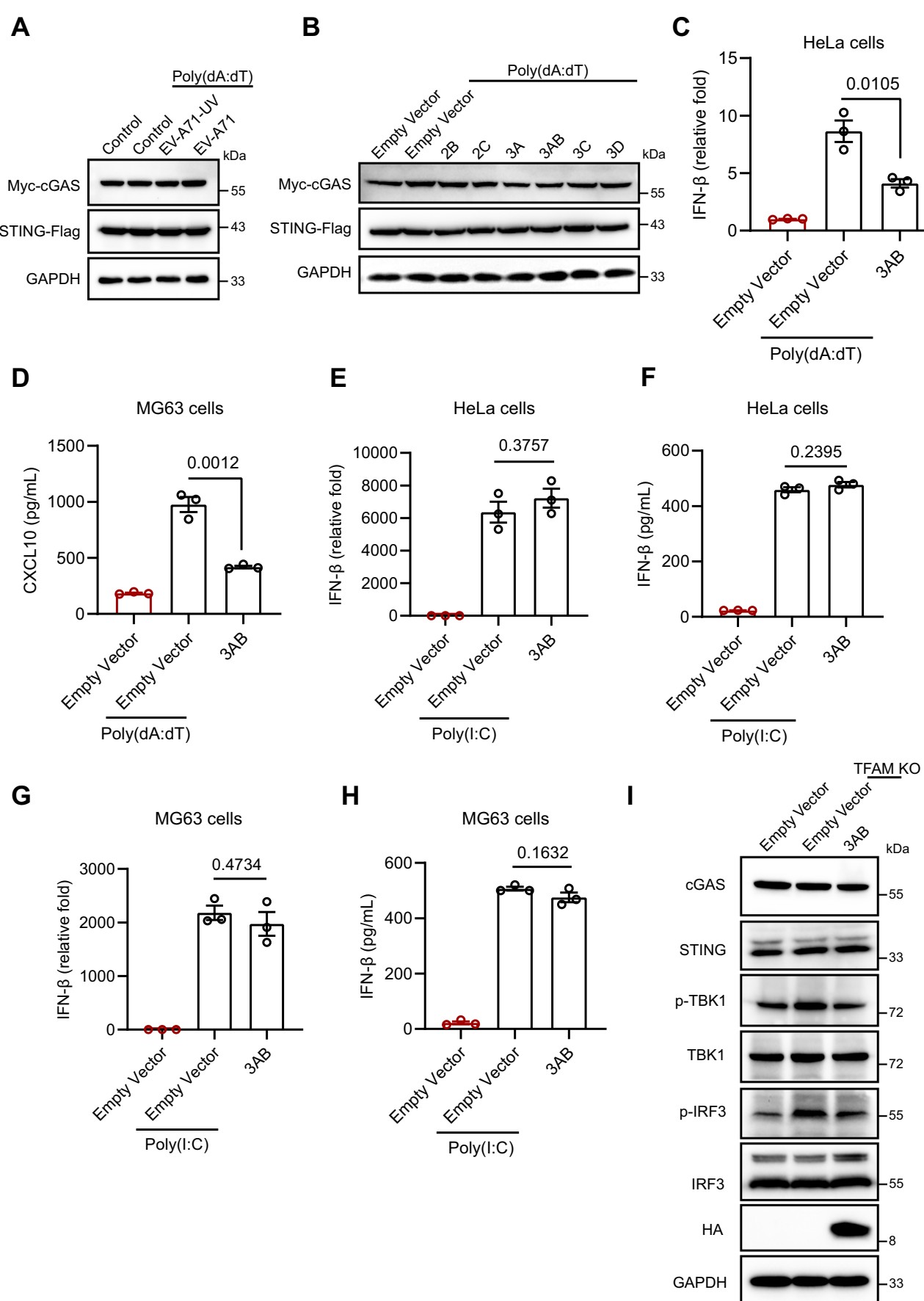

**Figure EV3. EV-A71 3AB inhibits DNA-mediated IFN-I responses.**

(A) cGAS, STING, poly(dA:dT), and IFN-β luciferase reporter plasmids were transfected into 293 T cells, which were then infected with EV-A71 or UV-inactivated EV-A71 for 12 h. The overexpression of cGAS and STING was assessed by Western blotting. (B) cGAS, STING, poly(dA:dT), and IFN-β luciferase reporter plasmids were transfected into 293T cells, which were then transfected with EV-A71 nonstructural plasmids for 24 h. The overexpression of cGAS and STING was assessed by Western blotting. (C) HeLa cells were transfected with 100 ng/mL poly(dA:dT) for 6 h, followed by transfection with EV-A71 3AB plasmid for 24 h. qRT–PCR was then performed to measure the expression of IFN-β mRNA. Data were presented as mean ± SEM, *n* = 3 biological replicates. Unpaired *t* test was used for statistical analysis. (D) MG63 cells were transfected with 100 ng/mL poly(dA:dT) for 6 h, followed by transfection with EV-A71 3AB plasmid for 18 h. CXCL10 protein expression in supernatant was determined by ELISA. Data were presented as mean ± SEM, *n* = 3 biological replicates. Unpaired *t* test was used for statistical analysis. (E, F) HeLa cells were transfected with EV-A71 3AB plasmid for 24 h, followed by transfection with 500 ng/mL poly(I:C) for 12 h. (E) The mRNA expression of IFN-β was assessed by qRT–PCR. (F) IFN-β protein expression in supernatant was determined by ELISA. Data were presented as mean ± SEM, *n* = 3 biological replicates. Unpaired *t* test was used for statistical analysis. (G, H) MG63 cells were transfected with EV-A71 3AB plasmid for 18 h, followed by transfection with 500 ng/mL poly(I:C) for 6 h. (G) The mRNA expression of IFN-β was assessed by qRT–PCR. (H) IFN-β protein expression in supernatant was determined by ELISA. Data were presented as mean ± SEM, *n* = 3 biological replicates. Unpaired *t* test was used for statistical analysis. (I) After 24 h of 3AB overexpression, Western blotting analysis was performed to assess cGAS, STING levels, as well as phosphorylated (p-) and total TBK1 and IRF3 levels, in TFAM knockout and mock cells. The experiments were repeated at least three times with the similar results. Source data are available online for this figure.

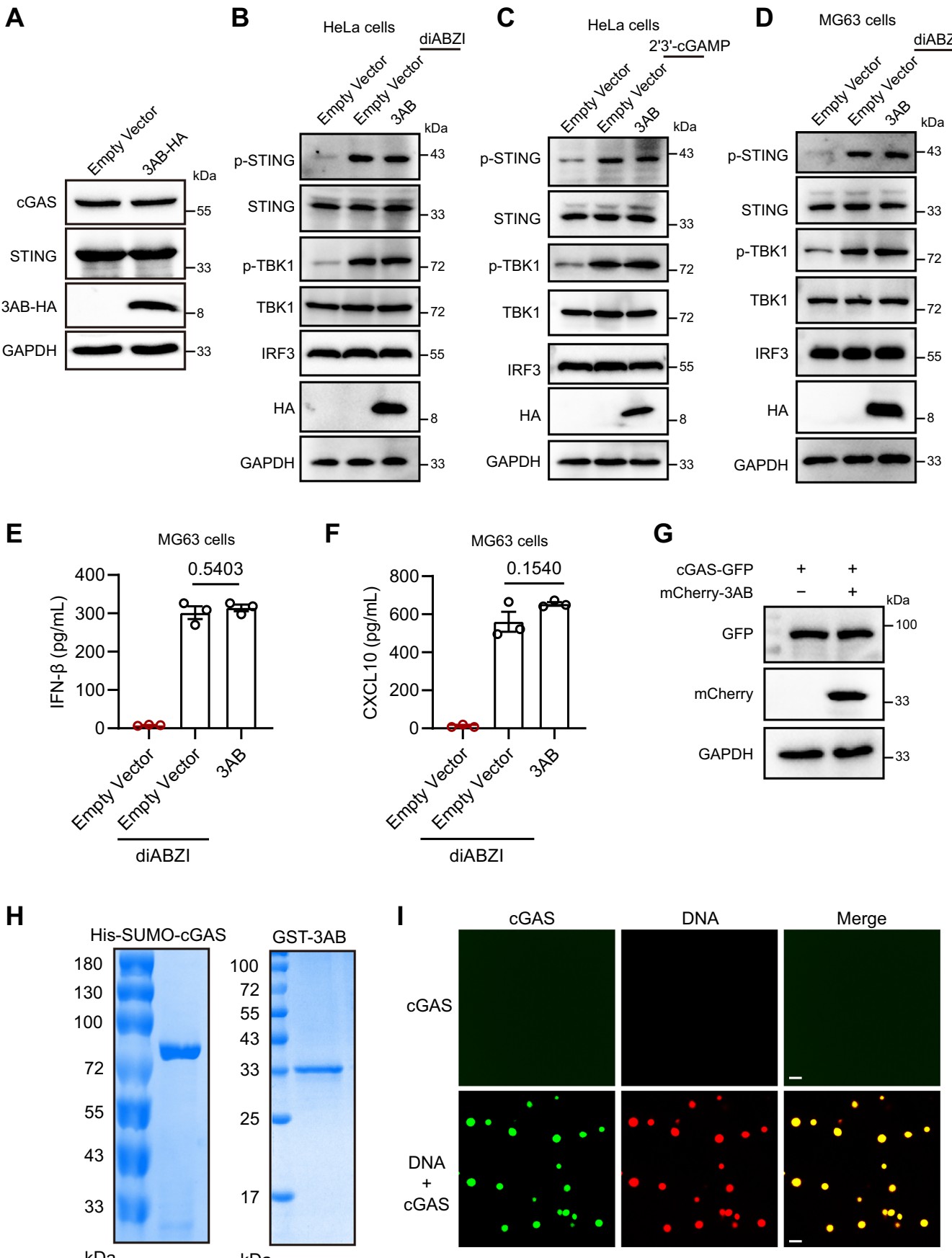

◀ **Figure EV4.   EV-A71 3AB does not inhibit STING agonist–induced innate immune responses.**

(A) After 24 h of 3AB overexpression, Western blotting analysis was performed to assess total cGAS, STING levels in HeLa cells. (B) The EV-A71 3AB plasmid was transfected into HeLa cells for 24 h, followed by treatment with diABZI (5 μM) for 4 h. Protein levels of IRF3, as well as total and phosphorylated TBK1 and STING, were analyzed by Western blotting. (C) HeLa cells were transfected with EV-A71 3AB plasmid for 24 h, followed by transfection with 2'3'-cGAMP (5 μg/mL) for 6 h. Protein levels of IRF3, as well as total and phosphorylated TBK1 and STING were analyzed by Western blotting. (D) The EV-A71 3AB plasmid was transfected into MG63 cells for 18 h, followed by treatment with diABZI (5 μM) for 4 h. Protein levels of IRF3, as well as total and phosphorylated TBK1 and STING, were analyzed by western blotting. (E, F) MG63 cells were transfected with the EV-A71 3AB plasmid for 18 h, followed by treatment with diABZI (5 μM) for 4 h. The protein levels of IFN-β (D) and CXCL10 (E) in supernatant were determined by ELISA. Data were presented as mean ± SEM, $n = 3$ biological replicates. Unpaired $t$ test was used for statistical analysis. (G) HeLa cells were co-transfected with cGAS-GFP, mCherry-3AB plasmids and mtDNA for 24 h. Western blotting analysis was performed to assess cGAS and 3AB using anti-GFP and anti-mCherry antibody. (H) Purification of the recombinant His-SUMO-cGAS and GST-3AB proteins produced in the *Escherichia coli* BL21 DE3 strain. The proteins were detected by Coomassie blue staining in SDS–PAGE gels. (I) Representative images of phase separation by mixing cGAS (20 μM) with 100-bp dsDNA (10 μM) in buffer containing 20 mM Tris-HCl (pH 7.5), 150 mM NaCl, 10 mM MgCl2 and 1 mg/mL BSA. Images were examined with a Leica STELLARIS 5 confocal microscope. Scale bar, 5 μm. The experiments were repeated at least three times with the similar results. Source data are available online for this figure.

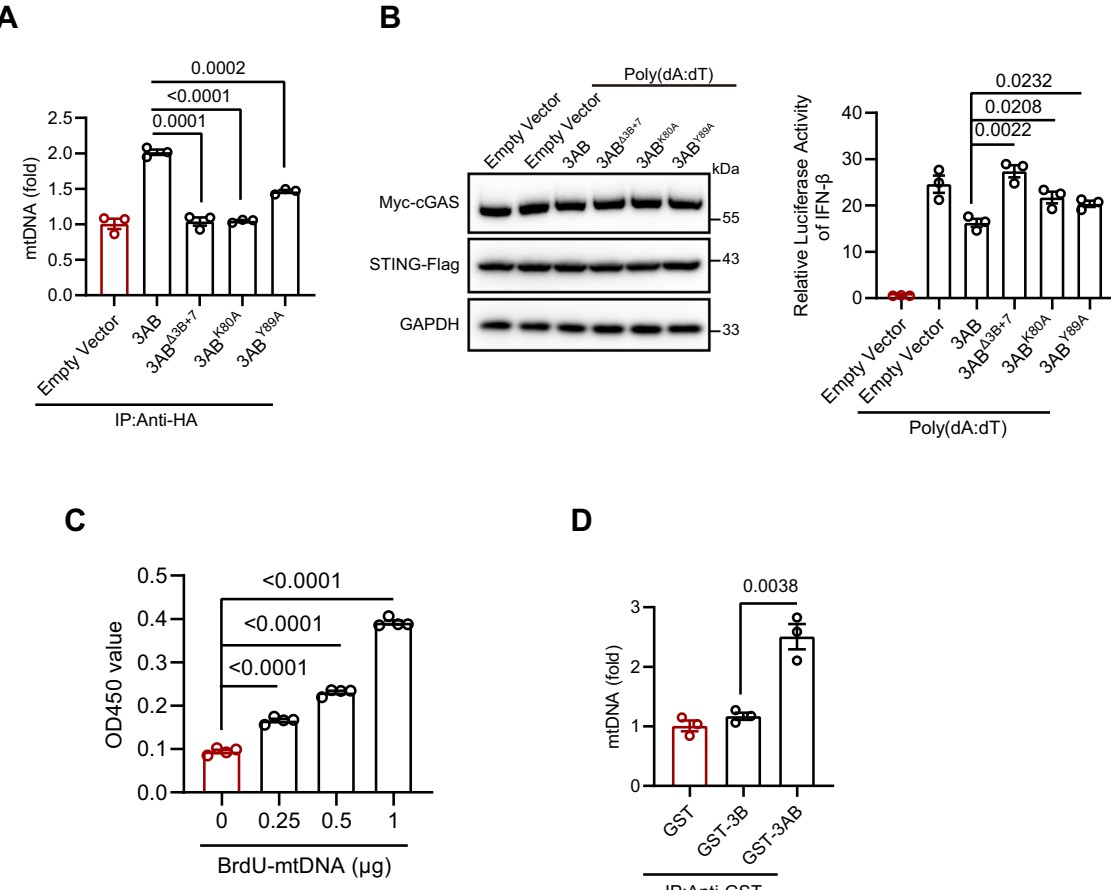

**Figure EV5.  EV-A71 3AB protein directly interacts with cytosolic mtDNA.**

(A) After 24 h of co-transfection with 3AB mutants plasmids and mtDNA, the cytosol was isolated, followed by immunoprecipitation with anti-HA. qPCR was performed to measure the cytosolic mtDNA levels. Data were presented as mean ± SEM, $n = 3$ biological replicates. Unpaired $t$ test was used for statistical analysis. (B) cGAS, STING, poly(dA:dT), and IFN-β luciferase reporter plasmids were transfected into 293 T cells, which were then transfected with EV-A71 mutants plasmids for 24 h. The overexpression of cGAS and STING was assessed by Western blotting, and IFN-β luciferase activity was measured with a dual-luciferase reporter assay. Data were presented as mean ± SEM, $n = 3$ biological replicates. Unpaired $t$ test was used for statistical analysis. (C) Purified GST-3AB protein was coated onto a 96-well ELISA plate and incubated overnight at 4 °C. BrdU-labeled mtDNA was then added and incubated for 2 h at room temperature. After washing, the bound mtDNA was detected using anti-BrdU antibodies. Data were presented as mean ± SEM, $n = 4$ biological replicates. Unpaired $t$ test was used for statistical analysis. (D) Purified GST-3AB protein and mtDNA were incubated together in binding buffer. The protein–mtDNA complexes were immunoprecipitated using anti-GST antibodies, and the associated mtDNA was quantified by qPCR. Data were presented as mean ± SEM, $n = 3$ biological replicates. Unpaired $t$ test was used for statistical analysis. The experiments were repeated at least three times with the similar results. Source data are available online for this figure.

