## [Peer Review File · EMBO Reports]

Enterovirus A71 3AB Protein Facilitates Immune Evasion by Blocking cGAS Recognition of mtDNA

Peng Sun, Xinya Yang, Jing Cui, Guicun Fang, Yuqin Wu, Jingyi Chang, Xiaofei Li, Yinli Xie, Lipeng Gan, Lina Ma, and Zhiyong Li

Corresponding author(s): Zhiyong Li (lizhiyong@wmu.edu.cn) , Peng Sun (sunpeng@wmu.edu.cn)

Review Timeline:

Submission Date:	27th Aug 25
Editorial Decision:	24th Oct 25
Revision Received:	20th Jan 26
Editorial Decision:	16th Feb 26
Revision Received:	22nd Feb 26
Accepted:	12th Mar 26

Transaction Report:

Dear Dr. Sun

Thank you for the submission of your research manuscript to our journal. I apologize for the delay in handling your manuscript, but we have now received the two enclosed reports on it.

As you will see, the referees acknowledge that the findings are interesting, but they also raise a number of concerns and have suggestions how to further strengthen the data. I think that all concerns are pertinent and should be addressed in a revision to strengthen the conclusion that EV-A71 3AB inhibits cGAS enzymatic activity.

Given these constructive comments, we would like to invite you to revise your manuscript with the understanding that the referee concerns (as detailed above and in their reports) must be fully addressed and their suggestions taken on board. Please address all referee concerns in a complete point-by-point response. Acceptance of the manuscript will depend on a positive outcome of a second round of review. It is EMBO Reports policy to allow a single round of revision only and acceptance or rejection of the manuscript will therefore depend on the completeness of your responses included in the next, final version of the manuscript.

We realize that it is difficult to revise to a specific deadline. In the interest of protecting the conceptual advance provided by the work, we recommend a revision within 3 months (January 24th, 2026). Please discuss the revision progress ahead of this time with the editor if you require more time to complete the revisions.

I am also happy to discuss the revision further via e-mail or a video call, if you wish.

=====
IMPORTANT NOTE:

We perform an initial quality control of all revised manuscripts before re-review. Your manuscript will FAIL this control and the handling will be delayed IN CASE the following APPLIES:

- 1) A data availability section providing access to data deposited in public databases is missing. If you have not deposited any data, please add a sentence to the data availability section that explains that.
- 2) Your manuscript contains statistics and error bars based on $n=2$. Please use scatter blots in these cases. No statistics should be calculated if $n=2$.

=====
When submitting your revised manuscript, we will require:

- 1) a .docx formatted version of the manuscript text (including legends for main figures, EV figures and tables). Please make sure that the changes are highlighted to be clearly visible.
- 2) individual production quality figure files as .eps, .tif, .jpg (one file per figure). Please download our Figure Preparation Guidelines (figure preparation pdf) from our Author Guidelines pages <https://www.embopress.org/page/journal/14693178/authorguide> for more info on how to prepare your figures.
- 3) a .docx formatted letter INCLUDING the reviewers' reports and your detailed point-by-point responses to their comments. As part of the EMBO Press transparent editorial process, the point-by-point response is part of the Review Process File (RPF), which will be published alongside your paper.
- 4) a complete author checklist, which you can download from our author guidelines (<<https://www.embopress.org/page/journal/14693178/authorguide>>). Please insert information in the checklist that is also reflected in the manuscript. The completed author checklist will also be part of the RPF.
- 5) Please note that all corresponding authors are required to supply an ORCID ID for their name upon submission of a revised manuscript (<<https://orcid.org/>>). Please find instructions on how to link your ORCID ID to your account in our manuscript tracking system in our Author guidelines (<<https://www.embopress.org/page/journal/14693178/authorguide#authorshipguidelines>>)

6) We replaced Supplementary Information with Expanded View (EV) Figures and Tables that are collapsible/expandable online. A maximum of 5 EV Figures can be typeset. EV Figures should be cited as 'Figure EV1, Figure EV2' etc... in the text and their respective legends should be included in the main text after the legends of regular figures.

7) Before submitting your revision, primary datasets (and computer code, where appropriate) produced in this study need to be deposited in an appropriate public database (see <<https://www.embopress.org/page/journal/14693178/authorguide#dataavailability>>).

The accession numbers and database should be listed in a formal "Data Availability " section (placed after Materials & Method) that follows the model below (see also <<https://www.embopress.org/page/journal/14693178/authorguide#dataavailability>>). Please note that the Data Availability Section is restricted to new primary data that are part of this study.

Data availability

Additional information on source data and instruction on how to label the files are available <<https://www.embopress.org/page/journal/14693178/authorguide#sourcedata>>

10) Figure legends and data quantification:

- the name of the statistical test used to generate error bars and P values,
 - the EXACT p-values,
 - the number (n) of independent experiments (please specify technical or biological replicates) underlying each data point,
 - the nature of the bars and error bars (s.d., s.e.m.)
-
- If the data are obtained from n {less than or equal to} 5, show the individual data points in addition to the SD or SEM.
 - If the data are obtained from n {less than or equal to} 2, use scatter blots showing the individual data points.

11) Our journal encourages inclusion of *data citations in the reference list* to directly cite datasets that were re-used and obtained from public databases. Data citations in the article text are distinct from normal bibliographical citations and should

directly link to the database records from which the data can be accessed. In the main text, data citations are formatted as follows: "Data ref: Smith et al, 2001" or "Data ref: NCBI Sequence Read Archive PRJNA342805, 2017". In the Reference list, data citations must be labeled with "[DATASET]". A data reference must provide the database name, accession number/identifiers and a resolvable link to the landing page from which the data can be accessed at the end of the reference. Further instructions are available at <<https://www.embopress.org/page/journal/14693178/authorguide#referencesformat>>.

12) All Materials and Methods need to be described in the main text using our 'Structured Methods' format. According to this format, the Methods section includes a Reagents and Tools Table (listing key reagents, experimental models, software and relevant equipment and including their sources and relevant identifiers) followed by a Methods and Protocols section describing the methods, ideally using a step-by-step protocol format. The aim is to facilitate adoption of the methodologies across labs. Please download and fill our Reagents and Tools Table template (.docx), which you can find in our author guidelines: <https://www.embopress.org/page/journal/14693178/authorguide#structuredmethods>. When submitting your revised manuscript, please do not include the Reagents and Tools Table in the Methods section of the manuscript but upload it as a separate file choosing the file type "Reagent Table". An example of a Method paper with Structured Methods can be found here: <https://www.embopress.org/doi/10.15252/msb.20178071>.

13) As part of the EMBO publication's Transparent Editorial Process, EMBO Reports publishes online a Review Process File to accompany accepted manuscripts. This File will be published in conjunction with your paper and will include the referee reports, your point-by-point response and all pertinent correspondence relating to the manuscript.

Yours sincerely,

=====

Referee #1:

In the current manuscript, Sun et al. investigate how enterovirus A71 (EV-A71) modulate the cGAS-STING pathway in response to mitochondrial DNA (mtDNA) leaked into the cytoplasm. It should be noted that this topic was previously investigated by another lab/paper which made similar findings (<https://doi.org/10.1371/journal.ppat.1011132>)

The authors show that EV-A71 infection induces mitochondrial damage, leading to a small increase of mtDNA in the cytosol, aligning with the previous study. Although EV-A71 infection caused mtDNA release, the authors found the virus actively prevented sensing of cytosolic DNA - in agreement with the previous study suggesting that EV-A71 2C impacted TBK1 signalling. Notably, here the authors propose that EV-A71 3AB protein inhibits the DNA-mediated antiviral immune response through inhibition of cGAS sensing, through direct interaction/masking of cytosolic DNA - to prevent binding to cGAS. The paper therefore extends previous works, however adding an interesting new concept as to how the EV-A71 virus escapes cGAS sensing. Several points need to be further clarified to support the claims made.

Major comments:

1) The manuscript should be much more upfront about what was previously shown in the field - around the concept that EV-A71 and other picornaviruses lead to cGAS activation by cytosolic mtDNA release (through mPTP opening). Currently this reads as if they were first to make this observation and the above paper is only cited at the end of the manuscript. The abstract/introduction should be re-written clearly referencing what was previously shown in that paper. Figure 1 is mostly validating what was already shown and this should be clearly stated. Similarly Figure 2 find "surprising" that EV-A71 did not induce IFN β induction - but this

was inferred from the plos pathogen paper since EV-A71 2C protein impacted TBK1 signalling. Again, this needs to be corrected and presented in light of what is known.

2) One key assumption of the new finding of this paper is that cGAS enzymatic activity is inhibited by the viral proteins such as EV-A71 3AB. This really needs to be confirmed looking at the production level of cGAMP for instance in Figure 3C upon poly dA/dT transfection (and the same should be demonstrated for CVA9 and CVA-16 proteins).

3) The inhibition of the cGAS-STING pathway is primarily demonstrated through RT-qPCR analysis of IFN- β mRNA levels in a transient transfection model of cGAS agonists. Analyses of IP-10 and IFN- β levels by ELISA should be provided in THP-1 cells (or an alternative cell line expressing good levels of these cytokines such as MG63 cells) expressing the EV-A71 3AB protein. These new experiments should also include testing the effect of EV-A71 3AB on activation of STING using a small molecule agonist (such as diABZI) to support the specificity of the activity at the level of cGAS (given that EV-A71 2C protein inhibits TBK1).

4) The data from TFAM KO Hela cells assumes that cGAS-STING is activated to lead to IFN β activation. This should be confirmed using cGAS and STING small molecule antagonists (like H151 for STING).

5) The claim of an impact on liquid liquid phase separations is not substantiated sufficiently to be made here. In fact, it is unclear if the data shown in Figure 4F/G is using transfected mtDNA or speculates that endogenous mtDNA is at play (there is no information about the mtDNA source for this, or data actually showing there is mtDNA in the cGAS clusters seen - noting that overexpression of cGAS forms such condensates spontaneously), and the only depiction of aggregates shown is not sufficient. All that can be concluded is that cGAS puncta are smaller when 3AB is overexpressed (but this could just be due to a lower transfection efficiency of cGAS when co-transfected with 3AB - since there is no control showing the same amount of cGAS is expressed in both conditions). This claim should be removed from the abstract; to support it the authors would need to look at LLPS using recombinant cGAS incubated with recombinant 3AB protein, looking at formation of LLPS in vitro.

Minor comments:

- 1) The Materials and Methods section and figure legends lack sufficient detail for reproducibility. The authors should provide comprehensive information regarding reagents, experimental conditions, and analysis techniques.
- 2) There is no mention of the source of CVA16 virus in the methods.
- 3) The authors mention protein transfection throughout the paper - presumably they mean transfection of plasmids overexpressing the proteins with lipo2000 as suggested in the methods? Please amend to mention the vectors are transfected.
- 4) The data shown states n=3 independent samples. Please clarify whether this is from 3 independent experiments (conducted on 3 different days) for each figure legend.
- 5) Please confirm that the RTqPCR used Actin as a housekeeping gene in the methods.
- 6) When the authors transfect purified mtDNA and show it activates cGAS, this is really using exogenous dsDNA - and should be made clear that this is not endogenous mtDNA.

Referee #2:

The manuscript reveals that EV-A71 2B protein induces mitochondrial damage upon infection, triggering the release of mitochondrial DNA (mtDNA) into the cytosol. Notably, the viral 3AB protein binds to and inhibits cGAS, disrupting its phase separation and enzymatic activity. Intriguingly, 3AB proteins from Coxsackievirus A (CVA) also demonstrate mtDNA-binding capability. While these findings are compelling, several critical aspects require clarification by the authors.

Major points

1. The authors should assess whether 2B protein overexpression alone induces mitochondrial damage and membrane potential changes (e.g., using JC-10 staining, TEM). This would confirm 2B's direct role independent of viral infection.
2. PolydA:dT can also activate RIG-I/MAVS through pol-III. The authors should utilize poly(I:C) (RIG-I/MAVS agonist) and HT-DNA (cGAS agonist) to demonstrate 3AB's selective inhibition of DNA sensing. The authors should treat cells with cGAMP (STING direct activator) or diABZI (STING small-molecule agonist). If 3AB does not suppress STING activation, its inhibitory effect is specific to cGAS.
3. The authors primarily rely on in vivo interaction experiments (IP or cell staining) to demonstrate EV-A71 3AB's inhibitory effect on cGAS. However, to conclusively establish the mechanism, purified proteins must be employed for rigorous biochemical validation. Previous studies have shown that viral proteins can disrupt DNA-mediated phase separation to suppress cGAS activity. Therefore, we recommend performing in vitro phase separation assays using purified 3AB protein, visualizing condensate formation via microscopy. Quantitative characterization of 3AB-DNA interactions should be conducted using surface

plasmon resonance (SPR) or microscale thermophoresis (MST). Additionally, cGAS enzymatic activity assays under varying 3AB concentrations would directly demonstrate inhibition kinetics. These complementary approaches would strengthen the mechanistic conclusions drawn from the cellular studies.

Minor points

1. Abbreviations: Define only at first use (e.g., DAMP, CVA, BrdU).
2. The authors should cite other viruses (e.g., HSV-1 VP22, KSHV ORF52, Sars-CoV2) that inhibit cGAS via competitive DNA binding, positioning 3AB's mechanism in context of discussion.

Dear Editor

We are grateful to the Editor and Reviewers for their insightful comments and constructive suggestions. We are pleased that the reviewers found our study interesting, compelling and important. As a result of extensive new work, the revised manuscript contains additional data that has now been added to both the main figures and supplemental figures. We firmly believe that the new data we have provided in response to the reviewers' suggestions have substantially contributed to the overall quality of our manuscript. We have taken all the constructive criticism to heart and addressed them in our detailed point-by-point response below. We hope that you find our revised manuscript suitable for publication in *EMBO reports*. The changed text in manuscript has been highlighted.

Comments from editor:

Thank you for the submission of your research manuscript to our journal. I apologize for the delay in handling your manuscript, but we have now received the two enclosed reports on it.

As you will see, the referees acknowledge that the findings are interesting, but they also raise a number of concerns and have suggestions how to further strengthen the data. I think that all concerns are pertinent and should be addressed in a revision to strengthen the conclusion that EV-A71 3AB inhibits cGAS enzymatic activity.

Given these constructive comments, we would like to invite you to revise your manuscript with the understanding that the referee concerns (as detailed above and in their reports) must be fully addressed and their suggestions taken on board. Please address all referee concerns in a complete point-by-point response. Acceptance of the manuscript will depend on a positive outcome of a second round of review. It is EMBO Reports policy to allow a single round of revision only and acceptance or rejection of the manuscript will therefore depend on the completeness of your responses included in the next, final version of the manuscript.

We realize that it is difficult to revise to a specific deadline. In the interest of protecting the conceptual advance provided by the work, we recommend a revision within 3 months (January 24th, 2026). Please discuss the revision progress ahead of this time with the editor if you require more time to complete the revisions.

I am also happy to discuss the revision further via e-mail or a video call, if you wish.

We thank the Editor for the constructive comments and valuable suggestions. To address the concern that EV-A71 3AB inhibits cGAS enzymatic activity, we performed a series of new biochemical experiments. Using purified proteins, we conducted *in vitro* phase separation assays and demonstrated that cGAS

undergoes robust DNA-induced phase separation, which is effectively disrupted by the addition of purified 3AB. Importantly, *in vitro* enzymatic activity assays showed that purified 3AB inhibited cGAS enzymatic activity in a dose-dependent manner.

Furthermore, surface plasmon resonance (SPR) analysis confirmed a direct and concentration-dependent interaction between 3AB and DNA.

To examine the specificity of 3AB-mediated inhibition, we performed stimulation experiments using cGAS or STING agonists. We found that 3AB selectively suppressed DNA-triggered cGAS signaling induced by HT-DNA or poly(dA:dT). In contrast, treatment with STING agonists (cGAMP or diABZI) showed that 3AB did not impair downstream STING activation, indicating that 3AB acts upstream of STING by targeting cGAS-dependent DNA sensing. We have addressed all of these points in detail in the following sections.

Referee #1:

In the current manuscript, Sun et al. investigate how enterovirus A71 (EV-A71) modulate the cGAS-STING pathway in response to mitochondrial DNA (mtDNA) leaked into the cytoplasm. It should be noted that this topic was previously investigated by another lab/paper which made similar findings (<https://doi.org/10.1371/journal.ppat.1011132>).

The authors show that EV-A71 infection induces mitochondrial damage, leading to a small increase of mtDNA in the cytosol, aligning with the previous study. Although EV-A71 infection caused mtDNA release, the authors found the virus actively prevented sensing of cytosolic DNA - in agreement with the previous study suggesting that EV-A71 2C impacted TBK1 signalling. Notably, here the authors propose that EV-A71 3AB protein inhibits the DNA-mediated antiviral immune response through inhibition of cGAS sensing, through direct interaction/masking of cytosolic DNA - to prevent binding to cGAS. The paper therefore extends previous works, however adding an interesting new concept as to how the EV-A71 virus escapes cGAS sensing. Several points need to be further clarified to support the claims made.

We thank the reviewer for finding our study interesting and extending previous findings. We have carefully addressed all the specific points raised by this reviewer in the following sections.

Major comments:

1) The manuscript should be much more upfront about what was previously shown in the field - around the concept that EV-A71 and other picornaviruses lead to cGAS activation by cytosolic mtDNA release (through mPTP opening). Currently this reads as if they were

first to make this observation and the above paper is only cited at the end of the manuscript. The abstract/introduction should be re-written clearly referencing what was previously shown in that paper. Figure 1 is mostly validating what was already shown and this should be clearly stated. Similarly Figure 2 find "surprising" that EV-A71 did not induce IFN β induction - but this was inferred from the plos pathogen paper since EV-A71 2C protein impacted TBK1 signalling. Again, this needs to be corrected and presented in light of what is known.

Response:

We thank the reviewer for this insightful comment. We have revised the Abstract and Introduction to clearly acknowledge the previous study (*Liu H et al., 2023, PLoS Pathogens, e1011132.*) demonstrating that EV-A71 and other picornaviruses induces mitochondrial damage and cytosolic mtDNA release (Line 23-24, Page 2; Line 74-76, Page 4). In the main text, we now explicitly state that the observations in Figure 1 are consistent with previous studies (Line 129, Page 6). In addition, we have removed the term “surprisingly” from the description of Figure 2.

2) One key assumption of the new finding of this paper is that cGAS enzymatic activity is inhibited by the viral proteins such as EV-A71 3AB. This really needs to be confirmed looking at the production level of cGAMP for instance in Figure 3C upon poly dA/dT transfection (and the same should be demonstrated for CVA9 and CVA-16 proteins).

Response:

We thank the reviewer for this important suggestion. Following the recommendation, we performed cGAS enzymatic activity assays by quantifying intracellular cGAMP levels after poly(dA:dT) stimulation. Our results show that EV-A71 infection markedly suppressed poly(dA:dT)-induced 2'3'-cGAMP production, whereas UV-inactivated EV-A71 did not inhibit cGAMP generation (Figure 3B). Importantly, expression of EV-A71 3AB significantly reduced intracellular cGAMP levels (Figure 4A), demonstrating that 3AB inhibits cGAS enzymatic activity. In addition, we performed the same cGAMP quantification assays for CVA9 and CVA16 3AB proteins. Consistently, both CVA9 and CVA16 3AB markedly suppressed cGAMP synthesis upon poly(dA:dT) stimulation (Figure 7F). These new data have been added to the revised manuscript (Line 187-191, Page 8; Line 196-197, Page 8; Line 228-230, Page 9; Line 332-333, Page 13).

3) The inhibition of the cGAS-STING pathway is primarily demonstrated through RT-

qPCR analysis of IFN- β mRNA levels in a transient transfection model of cGAS agonists. Analyses of IP-10 and IFN- β levels by ELISA should be provided in THP-1 cells (or an alternative cell line expressing good levels of these cytokines such as MG63 cells) expressing the EV-A71 3AB protein. These new experiments should also include testing the effect of EV-A71 3AB on activation of STING using a small molecule agonist (such as diABZI) to support the specificity of the activity at the level of cGAS (given that EV-A71 2C protein inhibits TBK1).

Response:

We thank the reviewer for this valuable suggestion. We performed ELISA measurements of IFN- β and CXCL10/IP-10 in MG63 cells expressing EV-A71 3AB. Consistent with our qRT-PCR results, 3AB expression markedly reduced IFN- β and IP-10 production upon poly(dA:dT) stimulation (Figure 3H and Figure EV3D). To further assess pathway specificity, we stimulated cells with the STING agonist diABZI. EV-A71 3AB did not inhibit diABZI-induced STING pathway activation, as assessed by Western blotting (Figure EV4D), nor did it affect downstream IFN- β and IP-10 production measured by ELISA (Figure EV4E–F). These results support the conclusion that 3AB specifically targets cGAS rather than STING. These new data have been added to the revised manuscript (Line 216-217, Page 9; Line 230-234, Page 9).

4) The data from TFAM KO HeLa cells assumes that cGAS-STING is activated to lead to IFN β activation. This should be confirmed using cGAS and STING small molecule antagonists (like H151 for STING).

Response:

We appreciate the reviewer's insightful suggestion. To confirm that the increased IFN- β activation observed in TFAM-KO HeLa cells is mediated through the cGAS–STING pathway, we performed additional experiments using both cGAS and STING small-molecule antagonists. Treatment with the cGAS antagonist RU.521 or the STING antagonist H151 significantly suppressed mtDNA-induced IFN- β production (Figure EV2E) as well as STING and TBK1 phosphorylation (Figure EV2F) in TFAM-KO cells, confirming that the enhanced interferon response is dependent on cGAS–STING signaling. These new results have been included in the revised manuscript (Line 172-175, Page 7).

5) The claim of an impact on liquid liquid phase separations is not substantiated sufficiently to be made here. In fact, it is unclear if the data shown in Figure 4F/G is using transfected mtDNA or speculates that endogenous mtDNA is at play (there is no

information about the mtDNA source for this, or data actually showing there is mtDNA in the cGAS clusters seen - noting that overexpression of cGAS forms such condensates spontaneously), and the only depiction of aggregates shown is not sufficient. All that can be concluded is that cGAS puncta are smaller when 3AB is overexpressed (but this could just be due to a lower transfection efficiency of cGAS when co-transfected with 3AB - since there is no control showing the same amount of cGAS is expressed in both conditions). This claim should be removed from the abstract; to support it the authors would need to look at LLPS using recombinant cGAS incubated with recombinant 3AB protein, looking at formation of LLPS *in vitro*.

Response:

We appreciate the reviewer's insightful suggestion. We now explicitly state in the revised manuscript that the condensates shown in Figure 4F-G were formed upon transfection of purified mtDNA, rather than endogenous mtDNA. This has been corrected and clarified in the figure legend (Line 838, Page 33).

To exclude the possibility that changes in condensate morphology result from altered cGAS expression, we performed Western blotting and confirmed that cGAS protein levels are similar in both the control and 3AB co-expression conditions (Figure EV4G) (Line 252, Page 10).

To further validate the effect of 3AB on cGAS–DNA phase separation, we conducted *in vitro* liquid–liquid phase separation (LLPS) assays using purified recombinant cGAS and 3AB proteins (Figure EV4H). We found that cGAS alone did not undergo phase separation in the absence of DNA *in vitro*, whereas the addition of DNA robustly induced cGAS-DNA condensate formation (Figure EV4I). Intriguingly, inclusion of purified 3AB protein disrupted the phase-separated cGAS–DNA condensates (Figure 4H). Importantly, *in vitro* enzymatic activity assays showed that purified 3AB inhibited cGAS enzymatic activity in a dose-dependent manner (Figure 4I). These results provide direct biochemical evidence that 3AB interferes with cGAS condensation. We have incorporated these findings into the revised manuscript (Line 254-260, Page 10).

Minor comments:

1) The Materials and Methods section and figure legends lack sufficient detail for reproducibility. The authors should provide comprehensive information regarding reagents, experimental conditions, and analysis techniques.

Response:

We thank the reviewer for this suggestion. We have added the necessary experimental details to the Materials and Methods section and figure legends.

2) There is no mention of the source of CVA16 virus in the methods.

Response:

We thank the reviewer for pointing this out. CVA16 virus was not used in this study, and only plasmid-expressed CVA16 3AB protein was analyzed.

3) The authors mention protein transfection throughout the paper - presumably they mean transfection of plasmids overexpressing the proteins with lipo2000 as suggested in the methods? Please amend to mention the vectors are transfected.

Response:

We thank the reviewer for this clarification. The term “protein transfection” refers to plasmid-based overexpression. To avoid confusion, we have revised the manuscript to use “plasmid transfection” throughout.

4) The data shown states n=3 independent samples. Please clarify whether this is from 3 independent experiments (conducted on 3 different days) for each figure legend.

Response:

The “n = 3” indicated in the figure legends refers to three biological replicates from the same experiment. We have now clarified this in the revised figure legends.

5) Please confirm that the RTqPCR used Actin as a housekeeping gene in the methods.

Response:

Actin was used as the housekeeping gene for normalization in qRT-PCR experiments. We have clarified this point in the Methods section (Line 512, Page 19).

6) When the authors transfect purified mtDNA and show it activates cGAS, this is really using exogenous dsDNA - and should be made clear that this is not endogenous mtDNA.

Response:

According to the reviewer's suggestion, we have clarified in the manuscript that the transfected mtDNA is exogenous.

Referee #2:

The manuscript reveals that EV-A71 2B protein induces mitochondrial damage upon infection, triggering the release of mitochondrial DNA (mtDNA) into the cytosol. Notably, the viral 3AB protein binds to and inhibits cGAS, disrupting its phase separation and enzymatic activity. Intriguingly, 3AB proteins from Coxsackievirus A (CVA) also demonstrate mtDNA-binding capability. While these findings are compelling, several critical aspects require clarification by the authors.

We thank the reviewer for recognizing the significance of our study and these findings are compelling. We have carefully addressed all specific points raised by the reviewer in the detailed responses below.

Major points

1. The authors should assess whether 2B protein overexpression alone induces mitochondrial damage and membrane potential changes (e.g., using JC-10 staining, TEM). This would confirm 2B's direct role independent of viral infection.

Response:

We thank the reviewer for this suggestion. We have performed experiments by overexpressing 2B in cells and evaluated mitochondrial damage and membrane potential using TEM and JC-10 staining. Consistent with our infection data, 2B overexpression alone is sufficient to induce mitochondrial damage (Figure EV1F) and reduce membrane potential (Figure EV1G), confirming its direct role independent of viral infection. These results have been added to the revised manuscript (Line 142-143, Page 6).

2. PolydA:dT can also activate RIG-I/MAVS through pol-III. The authors should utilize poly(I:C) (RIG-I/MAVS agonist) and HT-DNA (cGAS agonist) to demonstrate 3AB's selective inhibition of DNA sensing. The authors should treat cells with cGAMP (STING direct activator) or diABZI (STING small-molecule agonist). If 3AB does not suppress STING activation, its inhibitory effect is specific to cGAS.

Response:

We thank the reviewer for this insightful comment. We have now included experiments using poly(I:C) as a RIG-I/MAVS agonist and HT-DNA as a cGAS agonist to assess the specificity of 3AB. We found that 3AB markedly suppressed HT-DNA-induced IFN- β expression at both the mRNA and protein levels in HeLa cells, as determined by qRT-PCR and ELISA (Figure 3F-G).

Similarly, 3AB inhibited HT-DNA-induced IFN- β production in MG63 cells (Figure 3I-J). In contrast, 3AB did not significantly inhibit IFN- β expression induced by poly(I:C) (Figure EV3E-H) (Line 215-219, Page 9).

To further determine whether 3AB acts cGAS or STING, cells were treated with the STING agonists cGAMP or diABZI. In contrast to its inhibitory effect on DNA-triggered cGAS signaling, 3AB did not suppress cGAMP- or diABZI-induced activation of downstream STING signaling pathways (Figure EV4B-D), nor the expression of IFN- β (Figure EV4E-F). These results indicate that 3AB targets cGAS, rather than STING, to inhibit DNA sensing. These results have been incorporated into the revised manuscript (Line 230-235, Page 9).

3. The authors primarily rely on *in vivo* interaction experiments (IP or cell staining) to demonstrate EV-A71 3AB's inhibitory effect on cGAS. However, to conclusively establish the mechanism, purified proteins must be employed for rigorous biochemical validation. Previous studies have shown that viral proteins can disrupt DNA-mediated phase separation to suppress cGAS activity. Therefore, we recommend performing *in vitro* phase separation assays using purified 3AB protein, visualizing condensate formation via microscopy. Quantitative characterization of 3AB-DNA interactions should be conducted using surface plasmon resonance (SPR) or microscale thermophoresis (MST). Additionally, cGAS enzymatic activity assays under varying 3AB concentrations would directly demonstrate inhibition kinetics. These complementary approaches would strengthen the mechanistic conclusions drawn from the cellular studies.

Response:

We appreciate the reviewer's valuable suggestions. To provide biochemical evidence supporting the cellular findings, we expressed and purified recombinant cGAS and 3AB proteins for *in vitro* phase separation assays (Figure EV4H). We found that cGAS alone did not undergo phase separation in the absence of DNA, whereas the addition of DNA robustly induced cGAS condensate formation (Figure EV4I). Intriguingly, inclusion of purified 3AB protein disrupted the phase-separated cGAS–DNA condensates (Figure 4H). Importantly, *in vitro* enzymatic activity assays showed that purified 3AB inhibited cGAS enzymatic activity in a dose-dependent manner (Figure 4I) (Line 254-260, Page 10).

To further validate this interaction, we quantitatively analyzed the binding of 3AB to DNA at different concentrations using surface plasmon resonance (SPR), which confirmed a direct and dose-dependent 3AB–DNA interaction

(Figure 5G-H). These complementary biochemical data have been added to the revised manuscript (Line 291-293, Page 11).

Minor points

1. Abbreviations: Define only at first use (e.g., DAMP, CVA, BrdU).

Response:

All abbreviations, including DAMP, CVA, and BrdU, are now defined at their first appearance in the revised manuscript.

2. The authors should cite other viruses (e.g., HSV-1 VP22, KSHV ORF52, Sars-CoV2) that inhibit cGAS via competitive DNA binding, positioning 3AB's mechanism in context of discussion.

Response:

We have now cited studies on HSV-1 VP22, KSHV ORF52, and SARS-CoV-2 proteins that inhibit cGAS through competitive DNA binding (Line 372-374, Page 14).

Dear Dr. Li

Thank you for the submission of your revised manuscript to EMBO reports. We have now received the full set of referee reports that is copied below.

As you can see, both referees find that the study has been significantly improved during revision and recommend publication. Referee #1 comments on the statistical analysis. Please note that also our editorial policies mandate that statistics are only applied to data obtained from biological/independent replicates and may not be applied if the quantification is based on technical replicates or data obtained from one representative experiment. Please make sure that all quantification adheres to this policy.

Before I can accept the manuscript, I need you to address some minor points below:

- Please update the 'Conflict of interest' paragraph to our new 'Disclosure and competing interests statement'. For more information see <https://link.springer.com/journal/44319/submission-guidelines#conflictsofinterest>
 - Regarding the Author Contributions, we now use CRediT to specify the contributions of each author in the journal submission system. Therefore, please remove the Author Contributions from the manuscript file and make sure that the author contributions in our online manuscript tracking system are correct and up-to-date. The information you specified in the system will be automatically retrieved and typeset into the article. You can enter additional information in the free text box provided, if you wish.
 - The funding information in the manuscript text and online submission system must be congruent. Please add Ningbo Natural Science Foundation (2024J466), the Summit Advancement Disciplines of Zhejiang Province (Wenzhou Medical University-Pharmaceutics), and Science Technology Department of Wenzhou (2023S0094) in the system.
 - Please provide all figures as individual production quality Figure files.
 - Please provide weight markers for all Western blots.
 - Materials and methods should be Methods.
 - Reagents and Tools table: please remove the instructions paragraph from the file.
 - As a standard procedure we edit the title and abstract to make them more accessible to our general readership. Here, I only introduced a few minor changes. Please find my suggestion at the end of this e-mail.
 - Finally, EMBO Reports papers are accompanied online by
 - A) a short (1-2 sentences) summary of the findings and their significance,
 - B) 2-3 bullet points highlighting key results and
 - C) a schematic summary figure that provides a sketch of the major findings (not a data image).
 Please provide the summary figure as a separate file in PNG or JPG format at a size of 550x300-600 pixels (width x height). Please note that the size is rather small and that text needs to be readable at the final size. Please send us this information along with the revised manuscript.
- We look forward to seeing a final version of your manuscript as soon as possible.

With kind regards,

=====

Referee #1:

The authors have adequately addressed all the main concerns of this reviewer. One issue remaining is that since most figures only show representative data from independent experiments, and aggregate biological replicate from a single experiment, no statistics can be shown or referred to in the text (one cannot use stats on samples from a single experiment). This needs to be corrected before acceptance.

Referee #2:

The authors addressed all questions. The paper is ready to be accepted.

=====

Abstract

Mitochondrial DNA (mtDNA) is a powerful stimulator of the innate immune system and has been shown to trigger cytosolic DNA-sensing signaling during picornavirus infection. In this study, we observed that EV-A71 infection induces mitochondrial damage and leads to the release of mtDNA into the cytoplasm, which was mediated by the viral 2B protein. Despite this release, EV-A71 effectively suppresses the cGAS-STING-mediated type I interferon (IFN-I) response. We identify the nonstructural protein 3AB as a key viral antagonist of mtDNA sensing. Mechanistically, 3AB directly binds cytosolic mtDNA and disrupts cGAS-DNA phase separation, thereby suppressing cGAS-STING-dependent antiviral signaling. The immunosuppressive function of 3AB depends on the "3B+7" region, with mutations impairing its mtDNA binding and IFN-I suppression. Moreover, the 3AB proteins from coxsackievirus A9 (CVA9) and A16 (CVA16) also exhibit mtDNA-binding ability. This study reveals a novel immune evasion strategy by blocking mtDNA-triggered immune signaling, providing new insights into the interplay between viral infection and mitochondrial immune defense.

Comments from editor:

Thank you for the submission of your revised manuscript to *EMBO reports*. We have now received the full set of referee reports that is copied below.

As you can see, both referees find that the study has been significantly improved during revision and recommend publication. Referee #1 comments on the statistical analysis. Please note that also our editorial policies mandate that statistics are only applied to data obtained from biological/independent replicates and may not be applied if the quantification is based on technical replicates or data obtained from one representative experiment. Please make sure that all quantification adheres to this policy.

Response:

We thank the Editor and the referees for their positive evaluation of our revised manuscript and for recommending it for publication in *EMBO reports*. All statistical analyses presented in this manuscript are based exclusively on three or more independent biological replicates, in full compliance with the editorial policy of *EMBO reports*.

- Please update the 'Conflict of interest' paragraph to our new 'Disclosure and competing interests statement'. For more information see <https://link.springer.com/journal/44319/submission-guidelines#conflictofinterest>

Response:

We have revised the relevant section as requested.

- Regarding the Author Contributions, we now use CRediT to specify the contributions of each author in the journal submission system. Therefore, please remove the Author Contributions from the manuscript file and make sure that the author contributions in our online manuscript tracking system are correct and up-to-date. The information you specified in the system will be automatically retrieved and typeset into the article. You can enter additional information in the free text box provided, if you wish.

Response:

We have removed the Author contributions section from the manuscript file and use CRediT to specify the contributions of each author in the journal submission system.

- The funding information in the manuscript text and online submission system must be congruent. Please add Ningbo Natural Science Foundation (2024J466), the Summit Advancement Disciplines of Zhejiang Province (Wenzhou Medical University-Pharmaceutics), and Science Technology Department of Wenzhou (2023S0094) in the system.

Response:

We have added all the funding information in the online submission system.

- Please provide all figures as individual production quality Figure files.

Response:

We have provided all figures as individual production quality Figure files.

- Please provide weight markers for all Western blots.

Response:

We have provided weight markers for all Western blots.

- Materials and methods should be Methods.

Response:

We have revised the relevant section as requested.

- Reagents and Tools table: please remove the instructions paragraph from the file.

Response:

We have removed the instructions paragraph from the file.

- As a standard procedure we edit the title and abstract to make them more accessible to our general readership. Here, I only introduced a few minor changes. Please find my suggestion at the end of this e-mail.

Response:

We thank the Editor for these helpful suggestions and have incorporated them into the manuscript.

- Finally, EMBO Reports papers are accompanied online by
A) a short (1-2 sentences) summary of the findings and their significance,
B) 2-3 bullet points highlighting key results and
C) a schematic summary figure that provides a sketch of the major findings (not a data image).

Please provide the summary figure as a separate file in PNG or JPG format at a size of 550x300-600 pixels (width x height). Please note that the size is rather small and that text needs to be readable at the final size. Please send us this information along with the revised manuscript.

Response:

We have now added a short summary, 3 bullet points and a summary image as requested.

Enterovirus A71 infection triggers mitochondrial damage and mtDNA release while simultaneously suppressing mtDNA-driven innate immune signaling, revealing a viral immune evasion strategy targeting mitochondrial DNA

sensing.

- EV-A71 infection induces mitochondrial damage and cytosolic mtDNA release.
- The viral 3AB protein directly binds cytosolic mtDNA.
- 3AB-mediated disruption of cGAS–DNA phase separation suppresses type I interferon responses.

Referee #1:

The authors have adequately addressed all the main concerns of this reviewer. One issue remaining is that since most figures only show representative data from independent experiments, and aggregate biological replicate from a single experiment, no statistics can be shown or referred to in the text (one cannot use stats on samples from a single experiment). This needs to be corrected before acceptance.

Response:

We thank the reviewer for this comment and would like to clarify that all statistical analyses presented in the manuscript are based exclusively on three or more independent biological replicates, in accordance with the journal's policy. Thank you for your kind help during the peer review process.

Referee #2:

The authors addressed all questions. The paper is ready to be accepted.

Response: Thank you for your kind help during the peer review process.

Zhiyong Li
Wenzhou Medical University
Wenzhou Medical University
Zhejiang 325000
China

Dear Dr. Li,

I am very pleased to accept your manuscript for publication in the next available issue of EMBO reports. Thank you for your contribution to our journal.

You may qualify for financial assistance for your publication charges - either via a Springer Nature fully open access agreement or an EMBO initiative. Check your eligibility: <https://link.springer.com/journal/44319/how-to-publish-with-us>

Yours sincerely,

>>> Please note that it is EMBO Reports policy for the transcript of the editorial process (containing referee reports and your response letter) to be published as an online supplement to each paper. If you do NOT want this, you will need to inform the Editorial Office via email immediately. More information is available here: <https://link.springer.com/partners/embo-press/editorial-policies#Peer%20review>